# Automatic Understanding and Mapping of Regions in Cities Using Google Street View Images

**José Carlos Rangel** [1,2,3], **Edmanuel Cruz** [1,3,*] **and Miguel Cazorla** [4]

1   Grupo de Investigación RobotSIS, Universidad Tecnológica de Panamá (UTP), Centro Regional de Veraguas, Atalaya 0901, Panama; jose.rangel@utp.ac.pa
2   Facultad de Ingeniería de Sistemas Computacionales, Universidad Tecnológica de Panamá (UTP), El Dorado 0819, Panama
3   Sistema Nacional de Investigación (SNI) SENACYT, Panama City 0816, Panama
4   University Institute for Computer Research, University of Alicante, P.O. Box 99, 03080 Alicante, Spain; miguel.cazorla@ua.es
*   Correspondence: edmanuel.cruz@utp.ac.pa

**Abstract:** The use of semantic representations to achieve place understanding has been widely studied using indoor information. This kind of data can then be used for navigation, localization, and place identification using mobile devices. Nevertheless, applying this approach to outdoor data involves certain non-trivial procedures, such as gathering the information. This problem can be solved by using map APIs which allow images to be taken from the dataset captured to add to the map of a city. In this paper, we seek to leverage such APIs that collect images of city streets to generate a semantic representation of the city, built using a clustering algorithm and semantic descriptors. The main contribution of this work is to provide a new approach to generate a map with semantic information for each area of the city. The proposed method can automatically assign a semantic label for the cluster on the map. This method can be useful in smart cities and autonomous driving approaches due to the categorization of the zones in a city. The results show the robustness of the proposed pipeline and the advantages of using Google Street View images, semantic descriptors, and machine learning algorithms to generate semantic maps of outdoor places. These maps properly encode the zones existing in the selected city and are able to provide new zones between current ones.

**Keywords:** semantic maps; automatic map; outdoor understanding; deep learning

## 1. Introduction

People have always needed to know about the place they live in and the area within which they move, and have hence represented the known space in writing or through drawings. Maps are scale representations of a territory, which, when developed with metric properties, help measure different parameters with great accuracy. A semantic map is a kind of map that adds information about the main role of a place.

A map can provide useful data for different activities, such as the analysis of urban plans. Thanks to technological and scientific advances, everyone with Internet access can find free and interactive maps of all kinds, such as Google Maps and applications built over them, such as Google Street View (GSV). The latter is used in research such as [1], where it contributes to the design and evaluation of a new tool for the semiautomatic detection of ramps in GSV images using computer vision and machine learning. GSV has been used as a tool for others works, such as [2–4], as its utility for scene-understanding approaches has thus been demonstrated.

Many researchers have focused their attention on automatic map generation and outdoor scene understanding, using different techniques to perform these tasks. In this sense, Google Street View is currently one of the most important systems, providing us with easy visualization of an area.

Different map generation procedures currently focus on obtaining maps with several information levels, such as the semantic level [5–9]. These mapping procedures have been widely applied for indoor approaches using maps for robot navigation tasks [10–12]. Outdoor semantic map generation [13–16], on the other hand, produces fewer, but not worse results. Off-road paths in forests or non-urban areas are the primary focus of outdoor maps [13]. Nevertheless, urban regions will continue to be a challenge for these mapping procedures.

A common trend for generating semantic maps consists of using GSV [1–3] as an image source for the maps. Additionally, indoor or outdoor maps are built by applying supervised and unsupervised learning techniques, mainly convolutional neural networks [13,16–21] and clustering algorithms [5,22], respectively. However, outstanding results can be generated when using both approaches in conjunction [20,21,23,24]. It is worth noting that the semantic approach can today be viewed at two levels: at pixel level [20,25] and image level [23,26]. The first focuses on semantic maps for autonomous navigation, whereas the second is applied to navigation and localization within cities. The method proposed in this paper is based on the latter approach.

In recent years, semantic localization has been achieved using deep learning models and semantic descriptors [20,27–29]. These descriptors are generated using pretrained convolutional neural networks (CNN) models, such as those trained with the ImageNet dataset [30]. Each descriptor consists of a set of labels and their respective probability value, calculated by the CNN model. This kind of descriptor provides an additional description capability that allows for easy human understanding based on the meaning of the labels produced by the model. Then, applying an unsupervised learning algorithm, semantic groups are generated, based on the similarities provided by the labels.

The above methodology could be applied to outdoor scenarios. In this case, the places to be labeled involve residential areas, parks, or highways. In order to apply this concept to cities, a major drawback is the difficulty of finding a city dataset that contains images for the majority of the streets in the city. To overcome this problem, the present work will use images captured by GSV, and apply supervised (CNNs) and unsupervised (clustering) machine learning algorithms to create a semantic map that indicates the role of the different zones inside a town. This zone-labeling procedure will be addressed in an autonomous fashion, relying on the information provided. Our approach needs GPS information to obtain GSV images, and we then use other online and open access tools to obtain such information and thus create the experimental dataset.

The proposed system is able to build a semantic map of a given city. A semantic map of a city might have categories such as residential areas, alley, parking lot, or plaza. Our system is able to build the semantic map using third party images (GSV). In so doing, there is no need to take additional pictures from the city, and the process is conducted offline. Nonetheless, the proposed system depends on the GSV updates, i.e., it will show the city at the moment the pictures are taken from GSV.

The map can be easily updated when required.

The main contribution of this work is to provide a new approach to generate a map within semantic information for each area of the city using pictures of these regions. This semantic information is useful for driving and localization approaches in both operated and autonomous systems.

For example, when driving in an unknown city, the presence of semantics can be useful for planning a driving route or for the analysis of urban plans and smart city approaches. This information is also suitable for autonomous driving vehicles, allowing them to obtain data about regions of a city before entering their streets.

Furthermore, the method is semi-supervised: we do not define the categories a city has, but the method is able to obtain them using information from images.

The rest of the work is structured as follows: Section 2 presents the related works, while Section 3 describes our proposal. Section 4 explains how the maps will be generated and

describes the tools to do so. Sections 5 and 6 then explain the experiments and the results obtained, respectively. Finally, Section 7 outlines future work and concludes the paper.

## 2. Related Work

Maps are important for many applications, such as urban studies, tourism, and autonomous driving. Different techniques have been developed in recent years for the generation of maps applied to different fields, as described in [10], where the authors report that semantic mapping is a growing and active research topic. Below, we review various research works that perform map generation for different purposes.

The first work is presented by Li et al. in [31]. In this work, a method is proposed to derive detailed land-use information at the building block level, based on scene classification algorithms and GSV images. This study is interesting because land-use maps are important references for urban planning and studies. To implement this approach, the authors use GSV images and machine learning techniques, such as image feature descriptors and support vector machines (SVM).

The use of GSV images has also been tested for determining demographic levels in residential areas of the USA, as presented in [2], where the authors detect and analyze cars in the images. According to the authors, this proposed method will provide a new method of building-block-level land-use mapping. Additionally, Ruggieri et al. [3] used it as an image source for creating a dataset [4] for vulnerability analysis of buildings [3]. These works focus on images of a building in order to assign typological parameters to the building, and thus determine its vulnerability index.

The use of images in map generation procedures involves several sources of images; works such as [6,7] focus on taking aerial images to achieve semantic segmentation by applying convolutional neural networks (CNNs). These works produced maps considering only the top view of buildings. CNNs are also used for building hierarchical semantic maps to give a robot the capacity to interact with humans, using voice commands for indoor environments [18].

As is known, different methods and tools exist for map generation. Such is the case of Chen et al. [5], who construct high-quality city-scale maps from large collections of GPS trajectories, using graph-based clustering techniques with prior knowledge on road design. This allows for up-to-date and accurate digital maps. In the case of [32], authors use LiDAR sensors to detect the roads in the city streets with a semantic segmentation approach. As an external tool, the use of Global Information Service (GIS) for developing topological maps and navigation for urban robots is evaluated by Lee et al. in [33] for use with mobile robots.

Furthermore, the use of topological mapping is presented in studies such as [8,9,34–36]. These works involve the generation of the graph of the location and the construction of the map using different techniques, such as growing neural gas (GNG), SLAM, and analyzing the information provided by the sensors. These maps allow robots to understand, predict, and navigate inside the buildings.

Robot navigation has been improved by the use of several mechanisms for generating a semantic map of the environment. The authors in [11,12,36,37] generate semantic maps using 3D information with SLAM methods, sensors, and natural language descriptions, respectively. These maps allow the robots to move over determined areas. Semantic mapping has been studied in surveys [38,39], where the authors present a diversity of methods for generating semantic maps, mostly, however, applied to indoor environments. Consequently, several algorithms focus on solving semantic mapping for indoor locations, using robots. For outdoor scenarios, in contrast, the problem is not yet solved and the focus of several projects is to determine whether indoor methods might be applied in the same way for outdoor locations.

Over the last few years, the generation of semantic maps has focused mainly on the use of neural networks and other techniques with the aim of creating a label-based description for a set of images [17–19,21,40]. Among these works, the next level is represented by those that, in addition to the 2D information, use 3D sensors for gathering information on the

surroundings in order to generate a metric 3D map with semantic annotations [19,21,24]. This approach applies different matching procedures between 2D and 3D data to assign semantic labels produced by an image labeling model, to the points of a 3D cloud or capture. In this case, the result is pixel-level classification or a semantic segmentation process where every point of a 3D cloud will have a semantic label related to the RGB image labeling.

The main objective of semantic mapping procedures is the generation of a map that extensively represents the zone or area where the robot or vehicle will be navigating. In order to build maps, current works notably make use of [19], Kinect Fusion [21] and ORBSLAM2 [24].

The methodology mentioned above covers the more common working strategies for generating semantic maps for navigation tasks. It is worth mentioning that other research works apply a similar processing pipeline, but use distinct algorithms in each phase of the pipeline. In the semantic classification phase, we can find a greater diversity of techniques, although the most successful and popular are the CNN. The CNN architectures vary for each work. Some authors apply a pretrained classification model [17,18], while others use a modified version of some well-known convolutional architectures [13,16,19,20]. Specifically, we might mention YOLO [18], SSD [24], CNN [13,16,19,20], and a merging of RNN-CNN [21]. The use of these CNN models allows the generation of a semantic descriptor for labeling pixels, as presented in [17,20].

As another part of the pipeline, some works apply a data association phase [21,24]. This procedure focuses on assigning a semantic category, obtained by a classification model, to a specific cluster of points that have been detected in a $3D$ point cloud. In this way, it is possible to have an identified object in an RGB image, associated with the location of set of points existing in a 3D cloud. The main goal of this approach is to generate a semantic map based on the objects present and identified in an RGB image.

Using a similar processing pipeline, the work of Maturana et al. [13] focuses on outdoor scenarios. These authors applied segmentation algorithms and $2.5D$ grids maps to create semantic maps for autonomous navigation [13]. This work generates a map for semantically annotated elements in an image. The images consist of a view of an off-road path: the system assigns labels relevant to the navigation procedure, such as trail or grass. Images are captured with an all-terrain vehicle and 3D information is obtained from a LiDAR sensor, producing a map where each 3D point has a semantic label. Additionally, the work presented in [16] focuses on using micro-aerial vehicles for applying a semantic segmentation to an image capture taken from the vehicle. This approach follows a procedure but was designed for understanding images from an aerial viewpoint. This was designed for scouting purposes and to identify certain objects, such as cars, from the air.

In the same way, the research presented in [20] develops a topologic–semantic map for use in autonomous navigation in cities. This approach mainly seeks to assign a category to every pixel on an image using semantic segmentation with CNN models. Images are captured from the vehicle itself during the navigation and are labeled with a semantic descriptor of 4 dimensions/categories. The objective is to provide the vehicle with the knowledge of what is in its field of view to ease the navigation through the location. This approach does not use metric information.

As mentioned, map generation allows for multiple applications, such as the case of [22], which uses map generation to create maps for tourism use. To perform this task, the authors' system determines the salience of map elements, using bottom-up vision-based image analysis and top-down web-based information extraction techniques. The authors in [15] construct sound maps of urban areas, using social media data. This kind of map allows the authors to study the relationship between urban sounds and emotions. Another work, Ref. [14], develops CityMap, which is a system that generates transit maps from spatiotemporal crowd-sensed data collected via commuters' smartphones. One disadvantage of this system is that creating a city map is highly time-consuming.

In recent years, smart cities have become a common concept in various locations around the world. In these cities, the use of maps typically helps to solve a wide range of problems. Works such as [25,41,42] present solutions to accomplish the goal of smart cities. The solutions include an approach for the correct and efficient processing of the data. The semantic map helps understand the content of the cities and allows spatial maps of the cities to be built.

The approaches presented mainly focus on developing a semantic segmentation at a pixel level, whereas our proposal is to create a semantic map from an image level. This proposal seeks to assign labels at image level, based on the understanding of an image with CNN models. Each image will then have a category, and the clustering of the images will give an idea of the function or role of the zone within the city to which the image belongs.

As can be seen, different methodologies have been developed to generate maps for use in different tasks. However, the main difference between our proposal and previous works lies in their use of sensors or devices, such as lasers, GPS trajectory, etc., to acquire the information, whereas our proposal takes advantage of existing images available in Google Street View. Moreover, the desired semantic label assignation in our proposal is focused on images or scenes and not at pixel-level segmentation, as in the vast majority of existing works. In the following section, we detail the proposal of this work.

## 3. Proposal

In this work, we focus on generating a semantic representation of outdoor places using external tools and deep learning techniques. The objective is to build a map of a selected city using outdoor images. The map will be generated without human intervention and using open-access information. Images will be obtained with external APIs and then labeled using a pretrained CNN model to obtain a semantic descriptor [26]. These descriptors will be analyzed under an unsupervised bottom-up clustering algorithm and each generated cluster will include a set of semantically similar images that represents the function of the place within the city. The name or category of the semantic region will be automatically calculated based on the semantic descriptors that encode each image of the city dataset [23].

The generated map indicates the role of the different zones of the city and labels the city areas, such as highways, gardens, plazas, neighborhoods, etc., creating a semantic urban map.

Figure 1 shows the overall pipeline of the proposal, indicating how the outputs of the various parts interact and help generate the semantic map of the city. The complete procedure for the generation of the semantic urban maps could be summarized as follows:

- Dataset Gathering:
    - Determining the path between streets using the Google Map API (directions).
    - Computing the GPS coordinates within the paths. Note that the paths from the Google Map API provide a set of points in that path. If two of those points are too distant, we interpolate new points between them.
    - Gathering and storing the images belonging to the coordinates in the paths with the GSV API.

- City Information:
    - A graph of streets and intersections of a real city using the GeoApi.
    - A dataset of images of the coordinates in the streets.

- CNN Image Labeling:
    - Labeling the images obtained using a pretrained CNN model.

- Semantic Descriptors:
    - A set of semantic descriptors of all the images in the city dataset.

- Reduction of Semantic Descriptors:
    - Generating a mean semantic descriptor using the four semantic descriptors of each point in the graph.

- Mean Semantic Descriptors:
  - A set of mean semantic descriptors of all the points in the city graph.
- Semantic Map Generation:
  - Generating a semantic map using semantic descriptors and a similitude clustering procedure.
  - Smoothing the generated map with KNN.
- Semantic Map:
  - Final output consisting of a semantic map for the city graph, indicating the different zones within the city.

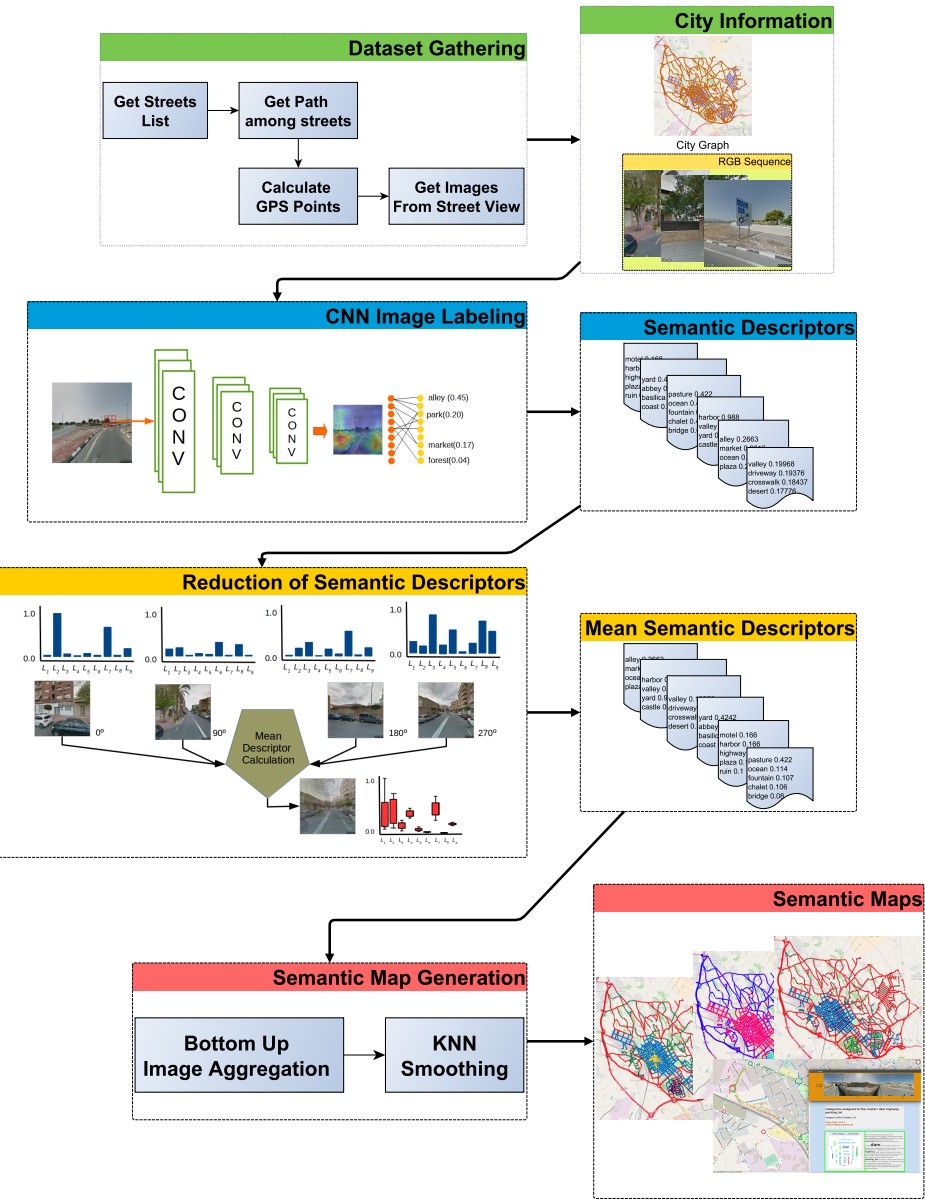

**Figure 1.** Proposal pipeline. This figure illustrates how the different procedures used in the proposal interact with the others and the output of each procedure.

## 4. Methodology

This section will present the different parts of the proposal and the tools used to achieve the information required to generate semantic maps about a city.

### 4.1. Gathering GPS Points

The proposal seeks to generate maps for outdoor scenes. Thus, all the images that will form part of the study need to belong to places located in a normal city; that is, they meet the common characteristics of most cities, such as streets, commercial and residential areas, parks, and industrial areas, among others.

To obtain the images, we will leverage several APIs that will allow us to gather different kinds of information about a specific city. Our approach is to automatically obtain the images from the Google Street View API. However, in order to gather the images, we need to define the set of GPS points for which we want the images. Hence, we rely on other APIs for this set of points, specifically, GeoAPI España and Google Maps Directions API.

#### 4.1.1. Geoapi España

GeoAPI España (https://geoapi.es/inicio (accessed on 2 February 2022)) is a service oriented towards developers. It helps normalize postal data in Spanish cities. Among the outputs generated by the API, we will use a list of streets generated by means of a city postal code.

This service is provided by the database of the Spanish National Institute of Statistics (INE) without cost. It is a public database used for the national census. The service requires a postal code from the city and then returns a list of the city's streets in a text format.

#### 4.1.2. Google Maps Directions API

Directions API (https://developers.google.com/maps/documentation/directions/start (accessed on 3 February 2022)) provides the directions for reaching a position on a map. This API provides not only the path, but also the GPS coordinates, at the point where the user has to turn, wait, or stop. It always provides the most efficient and shortest route between the start and end point. Figure 2 displays the indications generated for going to a particular place using the API. It allows users to indicate the means of transport, such as car, motorcycle, or bike, or we can even select walking. The outputs obtained by using this API are as follows:

- Routes between points.
- Selection of transportation method.
- Several Paths.

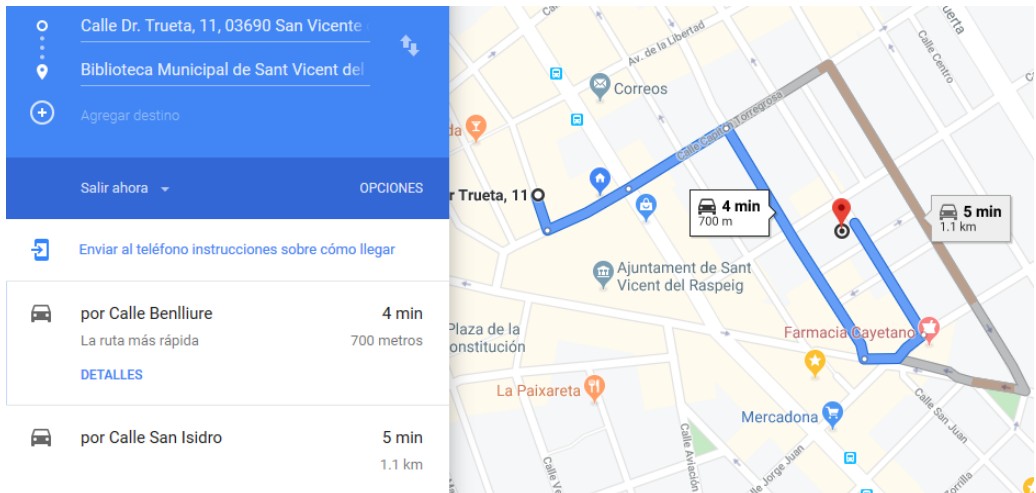

**Figure 2.** Routes API examples. This API returns a set of GPS coordinates on turning points and several intersections in the route. In this figure, these GPS points are marked with a white dot along the route.

### 4.2. Obtaining Images

To obtain the images for the dataset, we rely on Google Street View Static API for pictures of the city. Once all the GPS points of the destination city have been acquired, the Google Street View API will be used.

Google Street View Static API

Street View Static API (https://developers.google.com/maps/documentation/streetview/intro (accessed on 3 February 2022)) allows the user to obtain an image at a specific point, using a GPS coordinate. Requests can include a camera angle and tilt for the point (see Figure 3). Images come from the Google Street View database. This image database was captured outdoors, using a car or pedestrians with a special backpack camera device (for walking areas) in the cities. Images are then taken from a car view in a 360° and several tilt angles. Using this API, we are able to obtain the following:

- Images for a 360° view (headings or yaw).
- Several image sizes.
- Several camera orientations of the view (pitch).

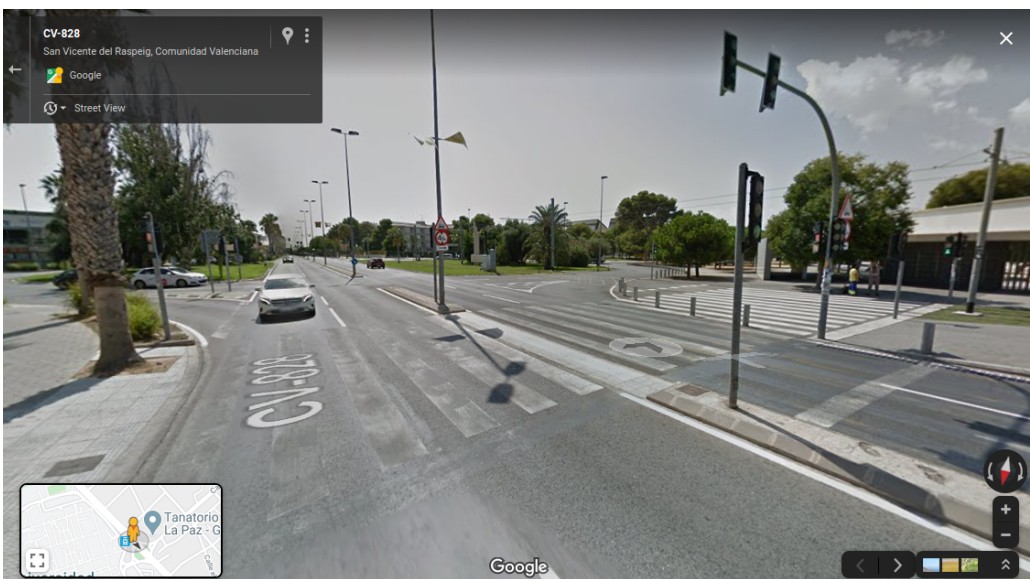

**Figure 3.** Google Street View image point of view.

### 4.3. GPS Point and Image Acquisition Procedure

To obtain the images to create the dataset for experimentation, we use the three APIs sequentially explained below. The first step is to select a city and obtain its list of streets, using GeoAPI España. Once the streets are obtained, we use this list and the Google Maps Directions API. These allow us to gather the instructions to travel between all the streets in the list, obtained in the former step. The instructions contain GPS coordinates that indicate the path to reach the target location. These GPS coordinates usually concern corners and intersections where the user should turn during the travel. Using these coordinates, we can obtain an initial graph with the points of the streets in the city.

The next step consists of using pairs of coordinates obtained with the Directions API. These coordinates are used to collect the GPS coordinates between them; in other words, to obtain the coordinates of places that are not corners or intersections and were not provided by the API. This procedure allows us to complete a GPS graph of the city with the information of several points inside a street. The following step consists of using these coordinates and the Google Street View Static API. This API lets us obtain an image from a given GPS position of the map, from the view of a car, with several angle options. Therefore, we request four images for every position in our city graph. We select pictures at

angles $\theta = 0°$, $90°$, $180°$, $270°$. These angles allow us to code the information surrounding a point or location in the city, as the use of a single image will omit these data. Figure 4 illustrates the procedure explained. After this step, we proceed to compute a semantic descriptor for every acquired image and to generate the semantic map, as will be explained in the following sections.

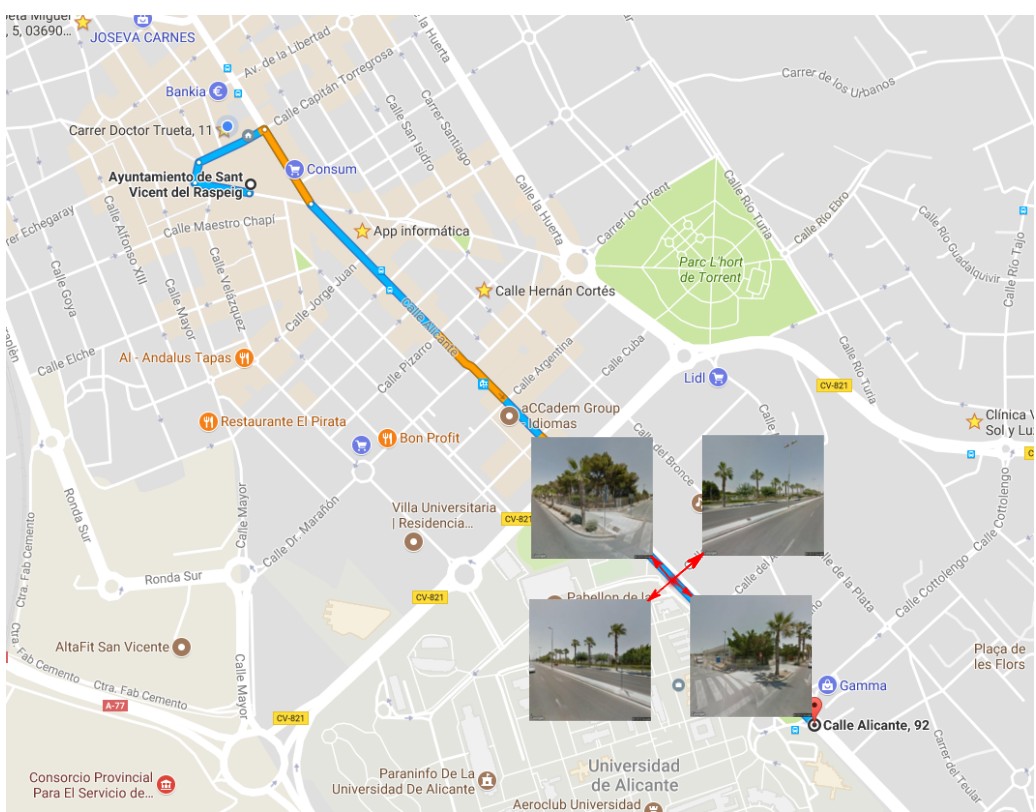

**Figure 4.** Example of images captured for every point on the map.

### 4.4. Lexical Labeling Using CNNs

4.4.1. Semantic CNN Descriptor

In this work, we are interested in the use of lexical annotations to automatically generate categories valid for semantic localization. This involves labeling each Street View capture, namely an RGB image, with a set of lexical terms. This process consists of using previously trained deep learning models, CNNs in our case, to assign a probability value to each lexical label that the model is able to recognize. This set of labels is formulated as in Equation (1):

$$\mathcal{L} = \{l_1, \ldots, l_{|\mathcal{L}|}\} \; where \; \sum_{i=0}^{|\mathcal{L}|} p_I(l_i) = 1. \tag{1}$$

where $\mathcal{L}$ is the set of predefined lexical labels of a CNN model, and $p_I(l_i)$ is the probability of describing an input image $I$ using the $i$-th label in $\mathcal{L}$.

This label assignation generates a lexical-based descriptor for every image analyzed by the CNN model. The generated descriptor can then be defined as in Equation (2):

$$d_{CNN}(I) = ([pI(l_1), \ldots, pI(l_{|\mathcal{L}|})]) \tag{2}$$

This produces an encoding that is similar to the bag of visual words [43], or the semantic manifold [44]. Based on this representation, each perception system of a mobile vehicle is expected to semantically describe its location at that specific moment [45]. Therefore, we can adopt lexical encoding to automatically categorize the locations of any outdoor environment from a semantic perspective. That is, two visually different images are assigned

to the same category if they represent similar lexical labels. To use a specific set of lexical descriptors, we need to select the pretrained CNN model.

### 4.4.2. Mean CNN Descriptor

The CNN descriptor encodes the features of a unique image/capture that belongs to a specific point on the map. In this proposal, for each coordinate on the map, we have a set of four images captured from the same GPS point but using a different angle in the camera. For each coordinate, we then define a mean descriptor that is computed, taking into account the individual descriptor for the four images of the point:

$$d_{CNN}(I_{lat,lon}) = ([mean(pI_{lat,lon\theta}(l_1)),$$

$$\ldots, \tag{3}$$

$$mean(pI_{lat,lon\theta}(l_{|\mathcal{L}|}))])$$

$$for\ \theta = 0°,\ 90°,\ 180°,\ 270°$$

where $pI_{lat,lon\ \theta}(l_i)$ denotes the probability of describing image $I$ at a given latitude and longitude with an angle $\theta$ in the camera orientation, using the $i$-th label in $\mathcal{L}$.

### 4.5. Bottom-Up Aggregation and Similarity Computation

The semantic mapping proposal follows a bottom-up aggregation process where sequences of RGB images serve as input. The visual information is exploited to extract a descriptor suitable for computing the similarity between two or more different captures, as proposed in [23]. This is achieved by following the procedure previously presented, which relies on the lexical annotations provided by deep learning procedures.

The bottom-up aggregation is carried out by means of a hierarchical clustering approach [46]. Initially, all the input images are established as different clusters, and every cluster is thus represented by the corresponding image descriptor ($d_{CNN}(I)$). These clusters are then combined iteratively, that is, the two most similar clusters are combined/aggregated in every new iteration. It is important to note that the final descriptor obtained for a specific cluster $C$ is equivalent to the element-wise average of the descriptors of all images belonging to this cluster $C$, that is,

$$d_{CNN}(C) = \left[ \frac{1}{|C|} \sum_{I \in C} p_I(l_1), \ldots, \frac{1}{|C|} \sum_{I \in C} p_I(l_{|L|}) \right] \tag{4}$$

The process stops when the most similar clusters are sufficiently different to be estimated as suitable to represent the same semantic category. This stopping condition can be tuned to select an optimal trade-off (threshold distance $\tau_d$) between specificity and generality. In this approach, the number of clusters will depend on the selected threshold, $\tau_d \propto \frac{1}{\#clusters}$. We discard the standard $k$-means clustering algorithms because the final number of clusters cannot be established beforehand. This clustering algorithm produces a dendrogram (Figure 5) that allows us to see how the clusters have been created and the distances between them.

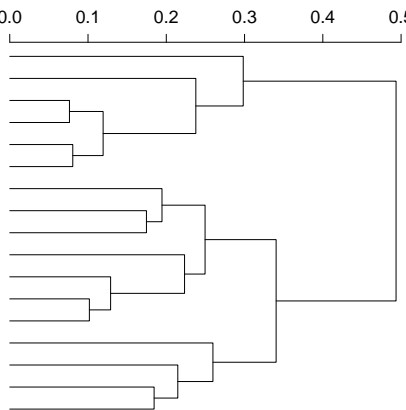

**Figure 5.** Example of hierarchical clustering procedure.

The selected bottom-up clustering approach is order-independent, which increases its usability, as no additional details from the source sequence of images, such as the velocity or the acquisition rate, are requested. The current implementation of the proposal includes a single combination per iteration involving the two most similar clusters.

Based on these preliminaries, the estimation of the cluster similarity is clearly established as the cornerstone of the proposal. As we adopt a representation where images are encoded using probabilities, we evaluate various similar measures that have been extensively studied [47]. The hierarchical clustering algorithm requires a set of parameters in order to generate the clusters. Table 1 shows the two main parameters of the method that will be evaluated in this study, the linkage strategy and the distance metrics, which are used to compute the distance between two clusters and define whether the image will merge a cluster or not.

A distance metrics equation is used to identify closeness between two points, two clusters, or a cluster and a point. This metric will determine how the points merge for creating the clusters. Linkage strategy is used to determine from where the distances between clusters will be measured.

**Table 1.** Parameters for the hierarchical clustering algorithm.

| Linkage Strategies | Distance Metrics |
|---|---|
| Average | Bray-Curtis |
| Centroid | Chebyshev |
| Complete | Correlation |
| Median | SqEuclidean |
| Single | |
| Ward | |
| Weighted | |

*4.6. KNN Map Smoothing Using GPS Coordinates*

Once the clusters have been assigned to each image using the semantic descriptor and the clustering procedure, we will apply an additional step to improve the quality of the map generated. As a result of the semantic mapping procedure, we obtain every GPS coordinate on the map linked to a cluster. As an expected result, the nearest points in a region of the map belong to the same cluster or semantic category. Nevertheless, in some cases, a unique point could be considered an outlier due to its having been assigned to a different cluster from the one that dominates the region.

Therefore, we take into account the position of every image on the map and the cluster assigned to the surrounding points. Every cluster assigned to the image/GPS position is then evaluated with those assigned to the nearest points using a KNN algorithm. Hence, the cluster assigned for each image is analyzed and changed if the majority of nearest points belongs to a different cluster.

## 5. Experimentation

### 5.1. Experimental Setup

In order to execute the experiments for the proposal, a set of images of the city under study is collected using the proposed procedure. The information on the city and the captured images is shown below.

- City: San Vicente del Raspeig, Alicante, España (see Figure 6).
- CNN Model: Places-GoogLeNet [48].
- Class or categories number: 205.
- Clustering threshold ($\tau_d$): $0.1 \rightarrow 1.0$.

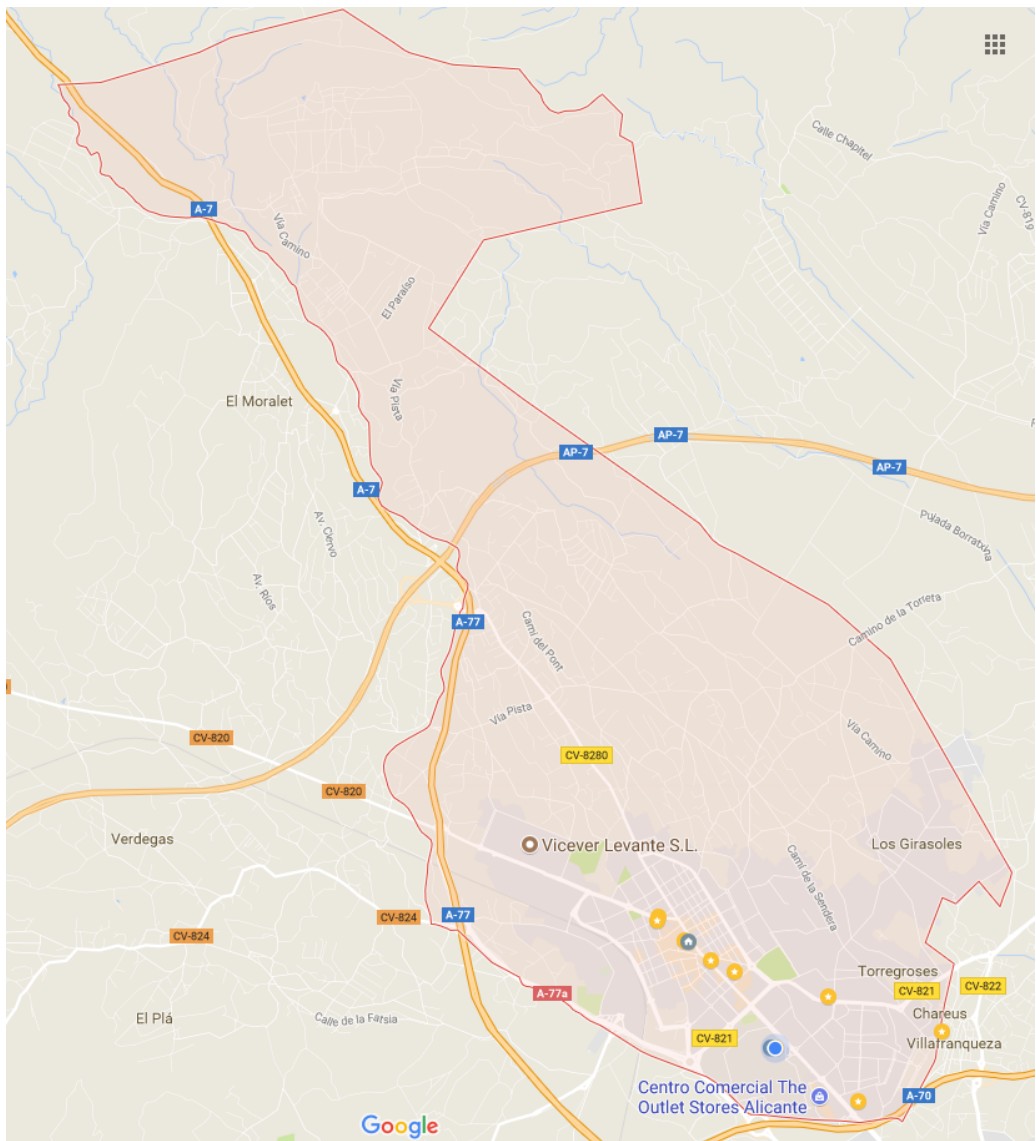

**Figure 6.** Map of San Vicente del Raspieg, the city used in this study (source: Google Maps).

The complete set of images was obtained using the Google Street View API and labeled using a pretrained CNN model. The Places dataset was chosen because it has the requirements to deploy the experiments. The number of classes refers to the number of categories in the dataset and, finally, the clustering threshold is the range investigated.

### 5.2. Pretrained CNN Model

Our experiments are carried out using the Places-GoogLeNet CNN-Model. This model is trained on the well-known Places MIT dataset [48]. This dataset was chosen because it contains images from 205 categories belonging to indoor and outdoor locations in cities. Table 2 shows some of the images included in the dataset and their respective label or category.

**Table 2.** Images and categories from the Places 205 dataset.

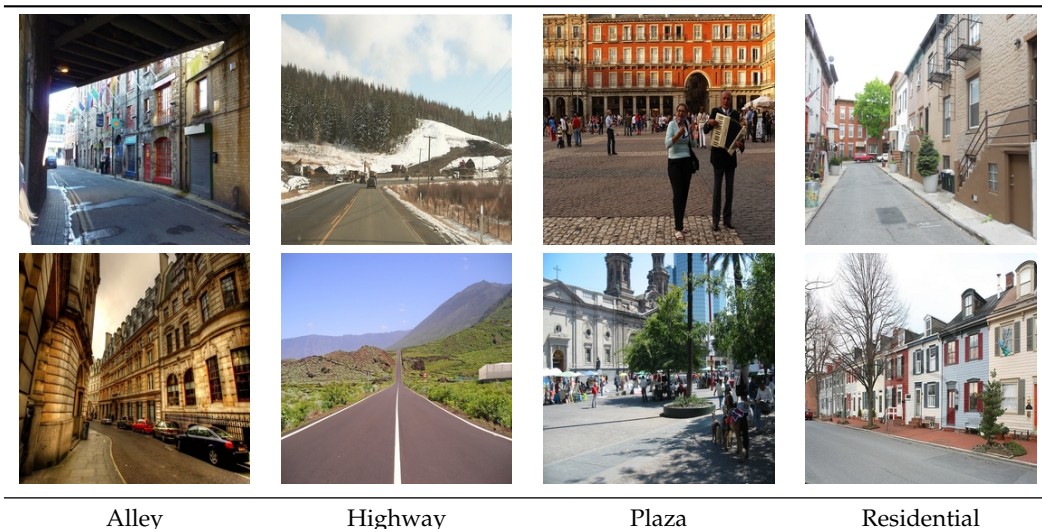

| Alley | Highway | Plaza | Residential |

### 5.3. Images from Street View API

The images for generating the semantic map were acquired using the Street View API. These images correspond to actual locations in the city selected for the study. The algorithm creates a graph with 24,185 points, producing a total of 96,740 images for use in the study. These images represent the city center and a great percentage of the streets inside and outside this center. Table 3 shows images obtained for several GPS points, at the four angles used in the study.

**Table 3.** Example images obtained with the Street View API.

Latitude = 38.37346 Longitude = −0.52512

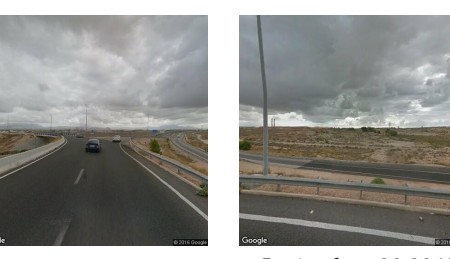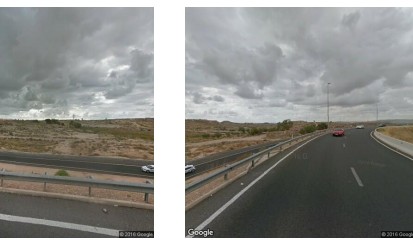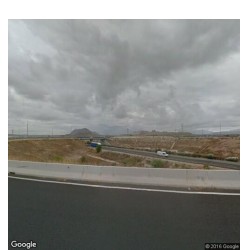
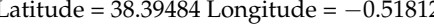

Latitude = 38.39484 Longitude = −0.51812

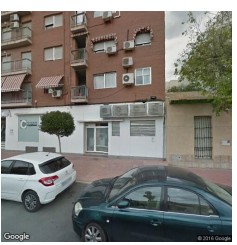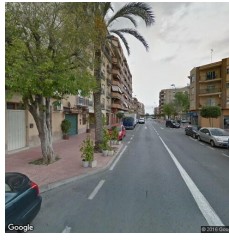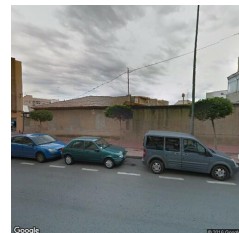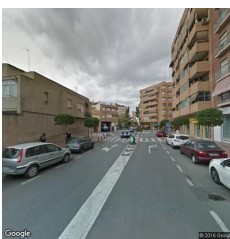

**Table 3.** *Cont.*

Latitude = 38.40601834 Longitude = −0.49542038

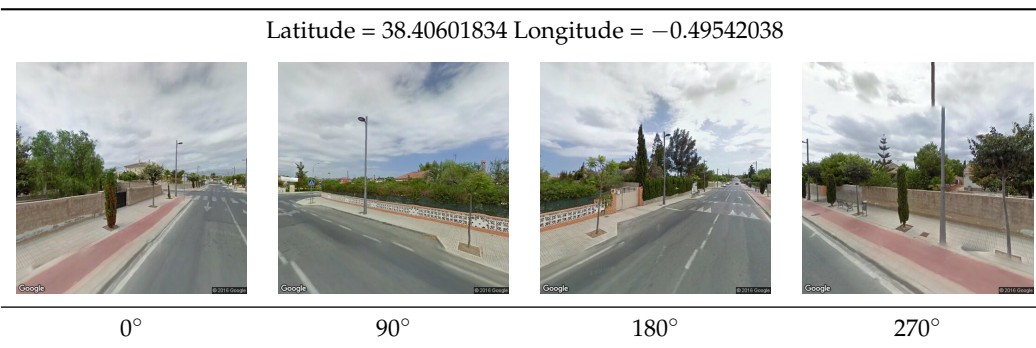

| 0° | 90° | 180° | 270° |

## 6. Results

### 6.1. Semantic Map Results

After executing the pipeline, a map is produced for different combinations of linkage strategy, distance metrics, and clustering threshold. To assess the results, it is necessary to qualitatively evaluate the different maps produced. Due to the large number of maps produced by the combination of parameters, we decided to only consider the maps that produced a number of clusters inside a defined range, based on the possible number of zones that can occur in the selected city. In this case, we set up this range from 5 to 30 to select a group of maps for evaluation.

Figure 7 shows the parameter combinations that were able to generate semantic maps with a number of clusters in the range of 5–30.

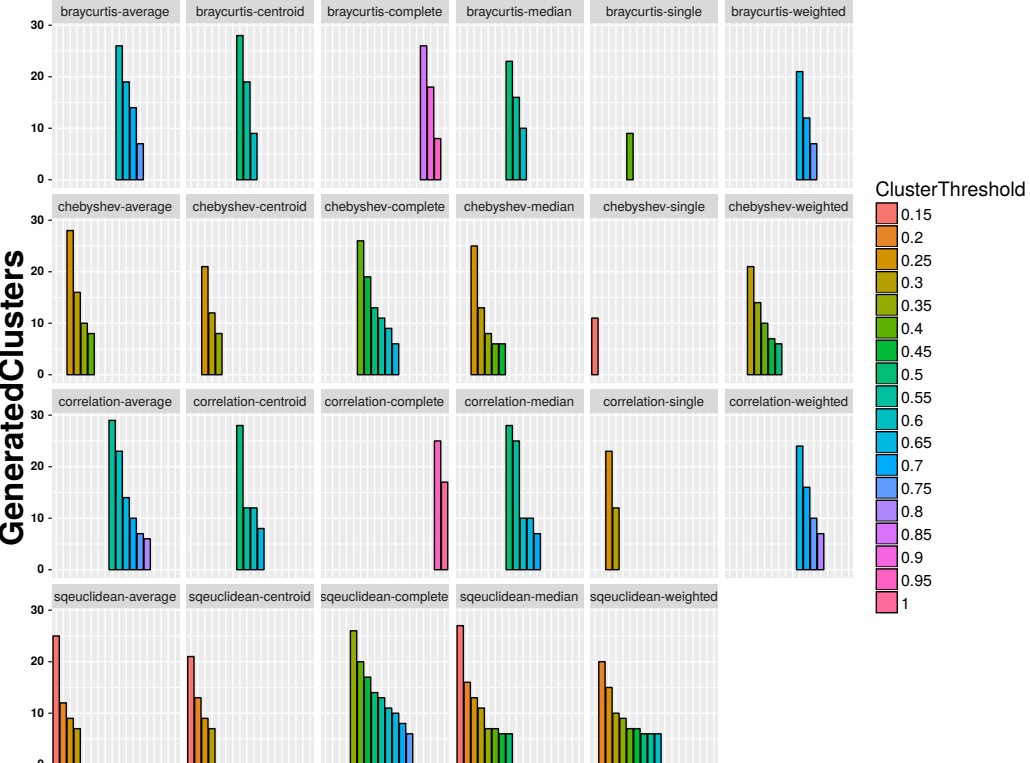

**Figure 7.** Each section of this chart shows the number of clusters generated by the combination of parameters in the algorithm, with every section representing a map. For each section, the x-axis indicates the threshold values used to cluster the map; each value is represented by a different color. The y-axis encodes the number of clusters generated by each threshold.

To decide which maps encode a good representation of the cities, it is necessary to study the maps that contain a number of clusters in the range defined below. In this case, we construct a histogram that represents the number of GPS points and the cluster; in other words, the number of clusters that contains a number of GPS points in a defined range.

A map that accurately represents a city must have an asymmetric distribution of the GPS points within the clusters of the map, avoiding the situation where only one cluster contains more than half the GPS points of the city. We have used the term density as a way to measure cluster distribution. In this case, density is the number of clusters with a given number of GPS points. In order to select the maps for evaluation, we determined that a histogram for a map that reliably describes the city must have the majority of its bars near the axis origin. This indicates that several clusters contain a smaller number of GPS points. In other words, we have a high concentration of clusters with a smaller number of GPS points. This distribution receives the name of positive asymmetric distribution.

Figure 8 shows the computed histogram (left) and the semantic map (right) represented by the histogram. The top of this figure shows a semantic map that poorly represents the city. The presence of a histogram bar near the right side of the chart means that several clusters contain almost all the points used in the study.

Figure 8 (bottom) shows a map that could be considered a good representation of the city, due to the clusters being mainly located on the left side of the chart.

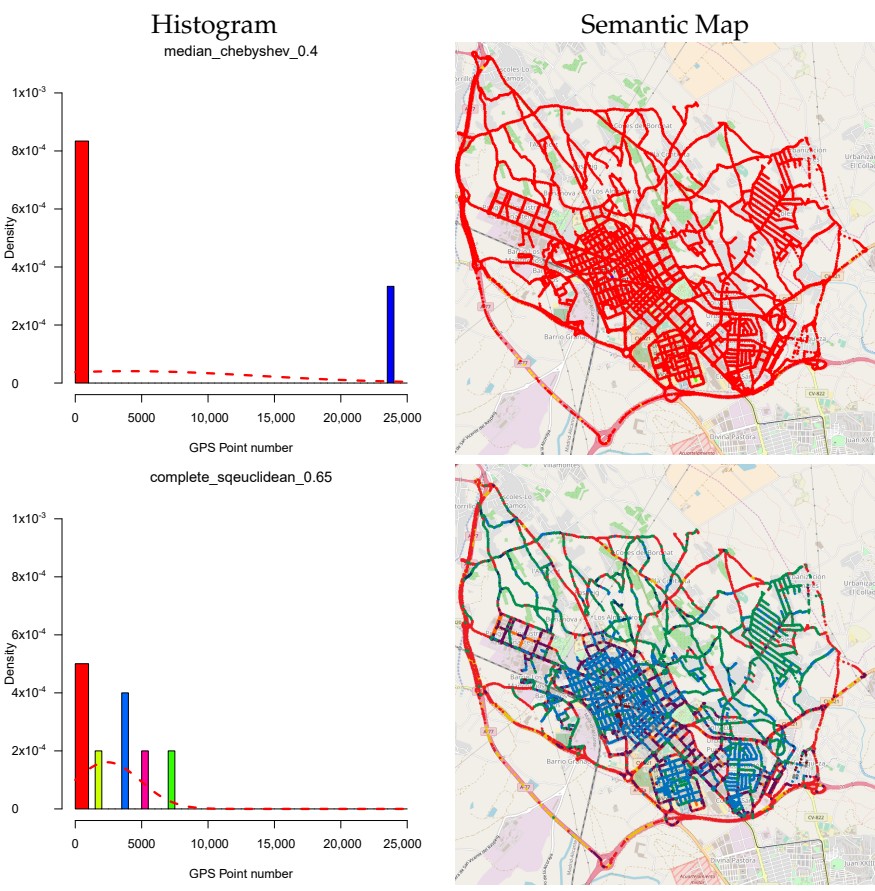

**Figure 8.** Density histogram (**left**) for semantic maps (**right**), generated using the proposed method. Histograms represent the density of clusters with the number of GPS points indicated in the *x*-axis. The higher number of bars on the left-hand side of the chart indicates a high number of clusters containing less than half the number of GPS points in the dataset. This may be understood as a feature of a good map. The cluster and bars colors indicate no relation between the concepts.

Of the maps that accomplish the proposed features, we selected that generated using Bray–Curtis as distance metric, average linkage strategy and a clustering threshold ($\tau$) of

0.6, hereafter called Average-BrayCurtis-0.6. The histogram for this map is presented in Figure 9, showing a high number of clusters composed of fewer than 5000 GPS points. This combination produces a total of 26 clusters, each corresponding to a semantic category assigned by the algorithm. Figure 10 (top) shows the map generated for the combination. The same figure (bottom) also presents a list with the clusters, the assigned color, main semantic categories, and the number of images/GPS points belonging to each cluster or semantic group.

**Figure 9.** Density histogram for combination Average-BrayCurtis-0.6. This histogram shows a high density of clusters containing fewer than 5000 GPS points, which indicates an asymmetric distribution of the GPS points between generated clusters.

The predominant lexical annotations in Figure 10 include three semantic labels with the highest mean probability value in the images belonging to the cluster. As shown in the figure, the cluster with most points featured the labels *highway parking_lot forest_road*. These are related to the highways, due to there being a vast number of images captured on the town's highways.

The histogram in Figure 11 indicates the occurrence frequency for a single semantic category within the categories assigned to the clusters. In this chart, the label *highway* has the highest frequency, pointing out that this label is present in the three highest probabilities of labels for 11 clusters. One of the reasons for the frequency of these labels is that the complete set of images of the city was captured from a car sight point. Streets and highways are then present in the vast majority of the images. Furthermore, the majority of the semantic categories used in this map correspond to outdoor categories of those available in the Places 205 pretrained CNN model. Therefore, the map contains semantic zones that represent the actual function or role that is developed in the location.

Figure 12 shows a capture of the information produced by the algorithm for the map. The black pin in the capture indicates the GPS point for which the information is shown. The window in the upper right-hand corner displays the GPS coordinate and the image acquired from Google Street View at this point. The bottom left-hand corner shows the semantic information obtained by the cluster in which the GPS point has been assigned by the algorithm. The word cloud represents labels with higher probability across all the images in the cluster. The greater the font size, the higher the probability is. This particular

point has been labeled by our algorithm as *alley*, *motel*, and *residential_neighborhood*, Then, by analyzing the images belonging to the point these correspond to, we find a one-way street in the city; the place includes residential buildings and commercial stores. Thus, the categories assigned by the algorithm represent the actual role of the location in the city.

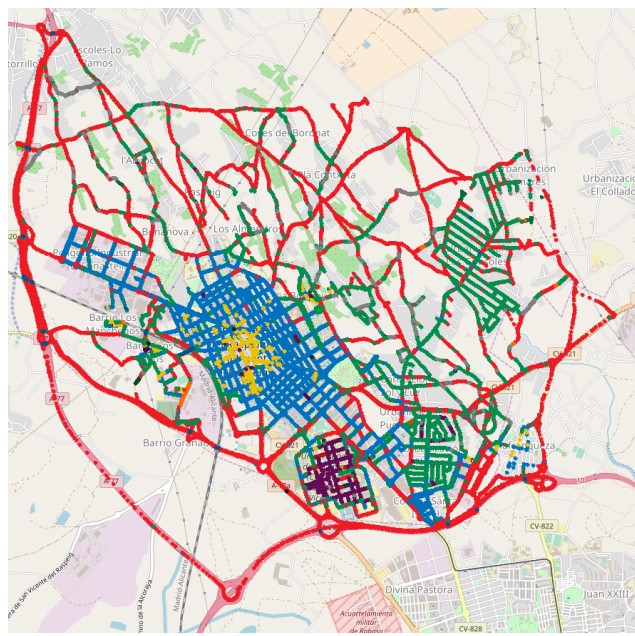

| Semantic Region Cluster | Predominant lexical annotations | #Images |
|---|---|---|
| | highway parking_lot forest_road | 10,253 |
| | parking_lot crosswalk highway | 8246 |
| | highway driveway parking_lot | 3985 |
| | driveway residential_neighborhood plaza | 542 |
| | orchard driveway highway | 408 |
| | alley motel residential_neighborhood | 363 |
| | railroad_track highway boardwalk | 75 |
| | plaza courtyard pavilion | 71 |
| | highway bridge basement | 63 |
| | desert_sand desert_vegetation highway | 62 |
| | train_station_platform residential_neighborhood railroad_track | 24 |
| | highway railroad_track forest_road | 23 |
| | botanical_garden formal_garden driveway | 16 |
| | dam highway parking_lot | 14 |
| | motel courtyard parking_lot | 12 |
| | airport_terminal plaza formal_garden | 6 |
| | wind_farm motel marsh | 6 |
| | slum train_station_platform motel | 4 |
| | shed corridor highway | 3 |
| | alley ruin driveway | 2 |
| | gas_station crosswalk highway | 2 |
| | crosswalk railroad_track picnic_area | 1 |
| | boardwalk playground crosswalk | 1 |
| | supermarket gift_shop shoe_shop | 1 |
| | corridor driveway veranda | 1 |
| | phone_booth lobby parking_lot | 1 |

**Figure 10.** Semantic clusters generated with the combination Average-BrayCurtis-0.6, producing 26 clusters.

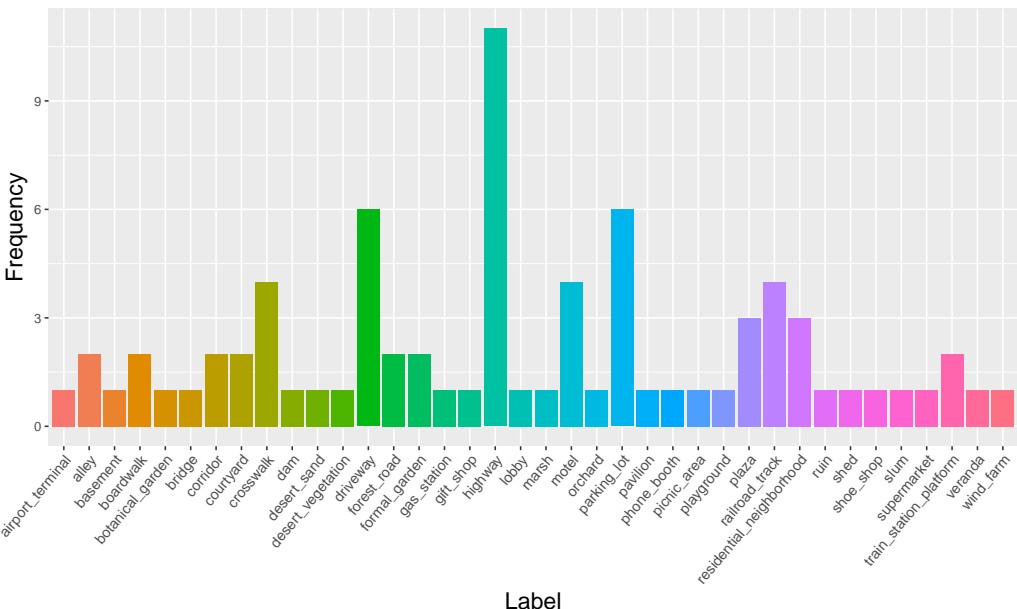

**Figure 11.** Label histogram produced with the Average-BrayCurtis-0.6 combination.

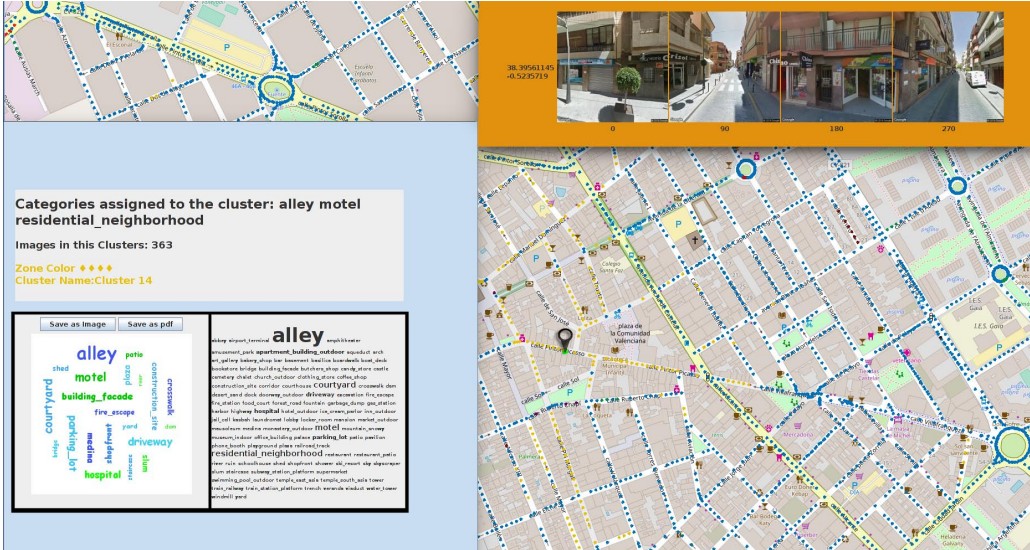

**Figure 12.** Details of the map generated for a specific point in the map.

Figure 13 displays the results for another point (marked with a black pin) on the map using the same parameter combination. Information on the cluster is shown in Figure 13 bottom. In this case, the point corresponds to images of a street that goes under a bridge. For this cluster, the predominant category is *highway*, and, for this example, the algorithm achieves the actual function of the location.

Figure 14 shows an example of a problem that could emerge during the creation of the semantic map. In this case, the high frequency is the label *dam* appearing in the images belonging to a cluster. The images also help to understand the reason for this high frequency, as these look highly similar to one that could be captured at a real dam.

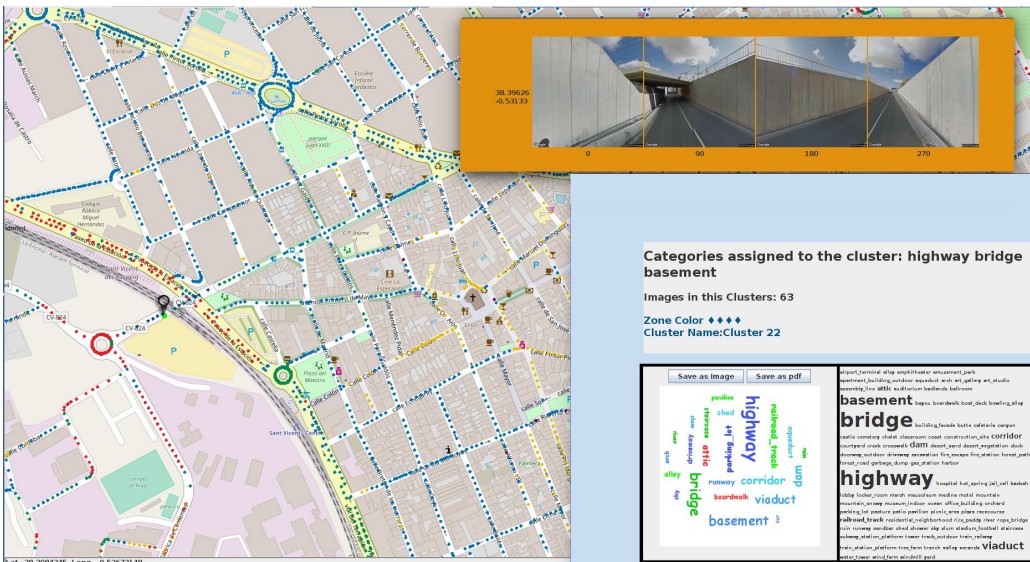

**Figure 13.** Details of the map generated for a specific point on the map.

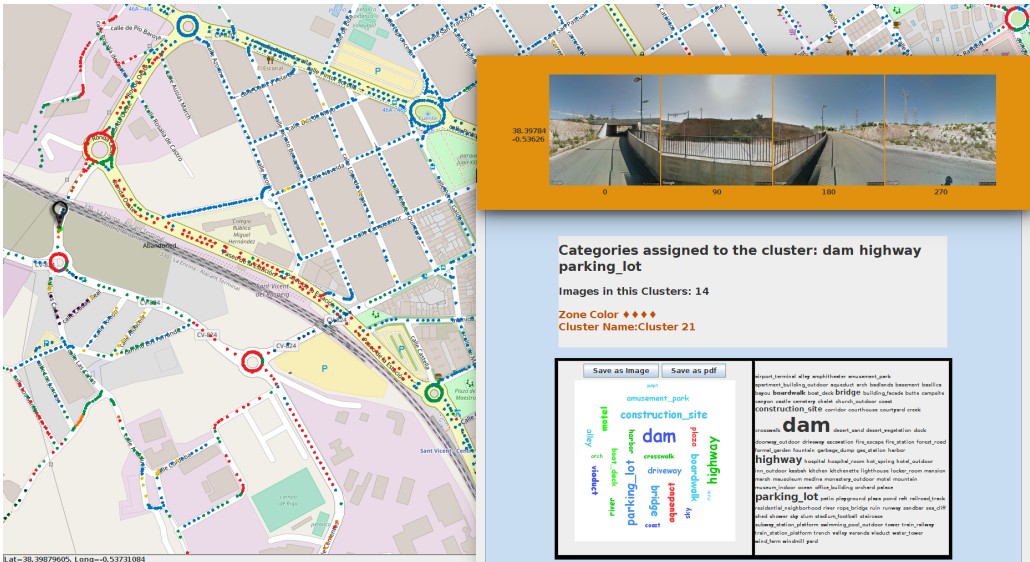

**Figure 14.** Failure case presented in the generated map; *dam* label occurrence in the middle of a field.

*6.2. KNN Smoothing Procedure*

In this section, we present the results obtained for the map generated with the Average-BrayCurtis-0.6 combination after applying the KNN smoothing procedure. The main objective is to reduce the number of clusters as well as reducing the outliers that could be present on the generated map.

This procedure uses the GPS coordinate and the assigned cluster for each point in the map. Then, by using a defined $k$ value , the clusters where the near points have been assigned are evaluated. If most neighbors belong to the same cluster, the label for the current point under evaluation remains the same; otherwise, the point is assigned to the cluster with more representatives in the $k$ selected points.

The KNN is applied with several $k$ values. Figure 15 (top) shows the result when using a $k = 15$ in the map presented in Figure 10 (top). In this case, the number of clusters after smoothing has been reduced from 26 to 15. Figure 15 (bottom) shows the clusters that remain after the smoothing procedure. Figure 16 shows how this procedure helps to reduce the number of semantic labels or zones present on the map.

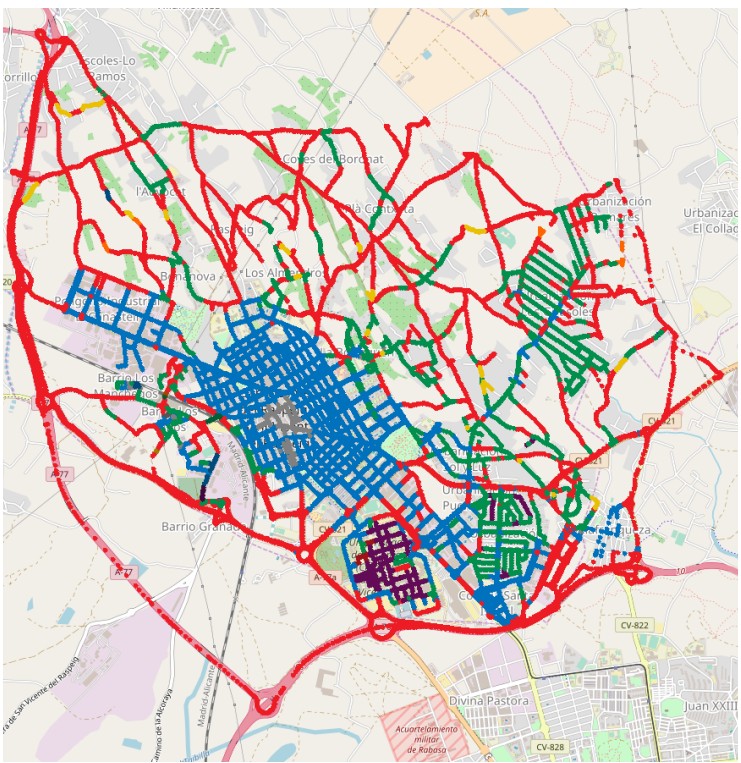

| Semantic Region Cluster | Predominant lexical annotations | #Images |
|---|---|---|
| | highway parking_lot forest_road | 10,741 |
| | parking_lot crosswalk highway | 8589 |
| | highway driveway parking_lot | 3689 |
| | driveway residential_neighborhood plaza | 511 |
| | alley motel courtyard | 230 |
| | orchard driveway highway | 217 |
| | desert_sand highway desert_vegetation | 62 |
| | plaza courtyard picnic_area | 49 |
| | highway railroad_track boardwalk | 48 |
| | highway bridge basement | 19 |
| | highway railroad_track forest_road | 14 |
| | train_station_platform boardwalk residential_neighborhood | 6 |
| | wind_farm motel desert_sand | 4 |
| | corridor basement dam | 4 |
| | airport_terminal lobby plaza | 2 |

**Figure 15.** Semantic map with Average-BrayCurtis-0.6 combination smoothed with $K = 15$, producing 15 clusters.

The histogram in Figure 16 shows that the number of categories present in the smoothed map were reduced. The remaining categories represent outdoor locations found in the city under study.

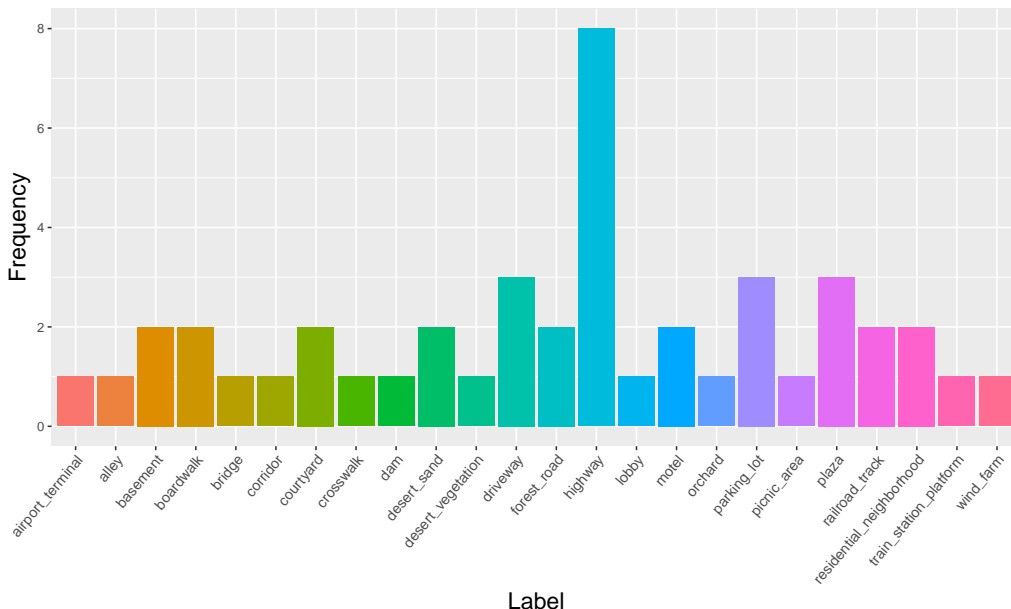

**Figure 16.** Label histogram produced with the Average-BrayCurtis-0.6 combination, smoothed with $K = 15$.

Figures 17 and 18 show the information generated by the algorithm for two points on the map. This data is calculated using the information from the smoothed map. The semantic category of each point is highly related to the appearance of the images belonging to them and also to the true role performed in the zone.

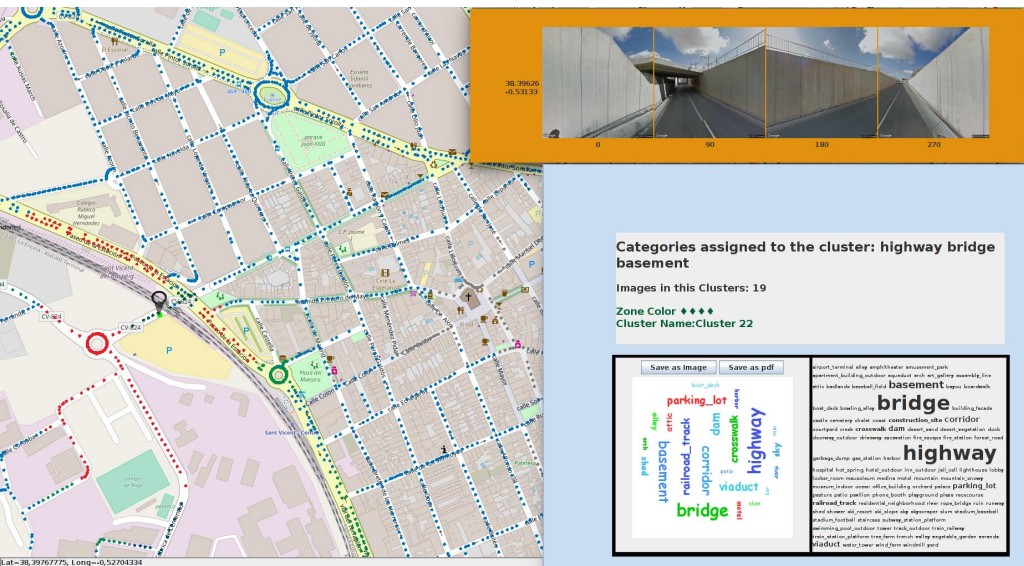

**Figure 17.** Example point in the map generated with the Average-BrayCurtis-0.6 combination, smoothed with $K = 15$.

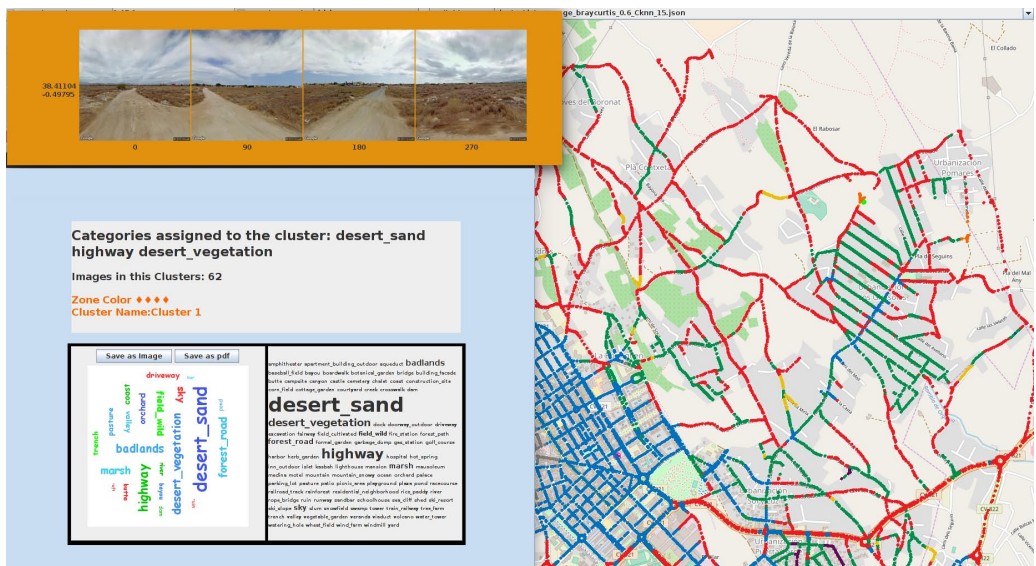

**Figure 18.** Example point in the map generated with the Average-BrayCurtis-0.6 combination, smoothed with $K = 15$.

Figure 19 shows the result of executing a smoothing procedure using a value of $k = 25$. In this case, the number of clusters has been reduced to 11 and the semantic categories continue representing actual zones of the city.

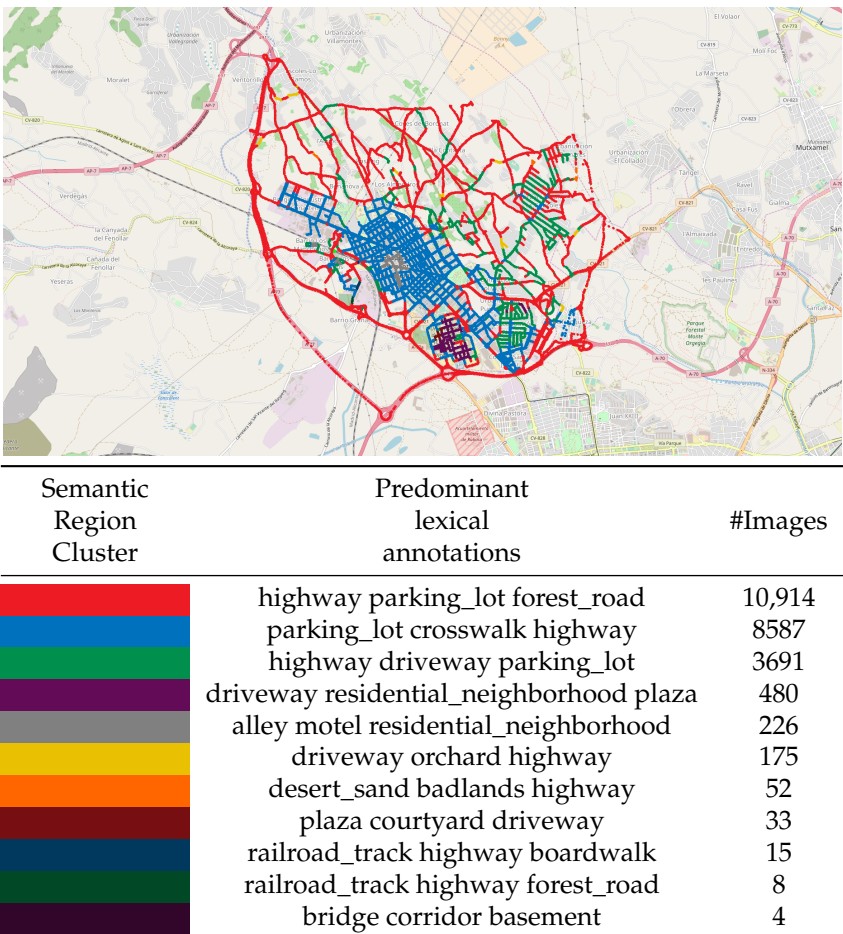

| Semantic Region Cluster | Predominant lexical annotations | #Images |
|---|---|---|
| | highway parking_lot forest_road | 10,914 |
| | parking_lot crosswalk highway | 8587 |
| | highway driveway parking_lot | 3691 |
| | driveway residential_neighborhood plaza | 480 |
| | alley motel residential_neighborhood | 226 |
| | driveway orchard highway | 175 |
| | desert_sand badlands highway | 52 |
| | plaza courtyard driveway | 33 |
| | railroad_track highway boardwalk | 15 |
| | railroad_track highway forest_road | 8 |
| | bridge corridor basement | 4 |

**Figure 19.** Semantic map with Average-BrayCurtis-0.6 combination smoothed with $K = 25$, producing 11 clusters.

### 6.3. KNN Smoothing Comparison

Figure 20 shows the map generated by the algorithm using the Average-BrayCurtis-0.6 combination (left) and the map obtained after applying the KNN smoothing procedure (right), using a value of $k = 15$. It is worth mentioning that several clusters remain on the new map and only change the originally assigned color. On the other hand, some clusters have disappeared from the map due to the number of points assigned to the cluster having decreased to zero. In this map, it can be seen that the larger the clusters, the greater the likelihood of remaining on the map is, while little clusters tend to be eliminated and their images or points added to a bigger cluster.

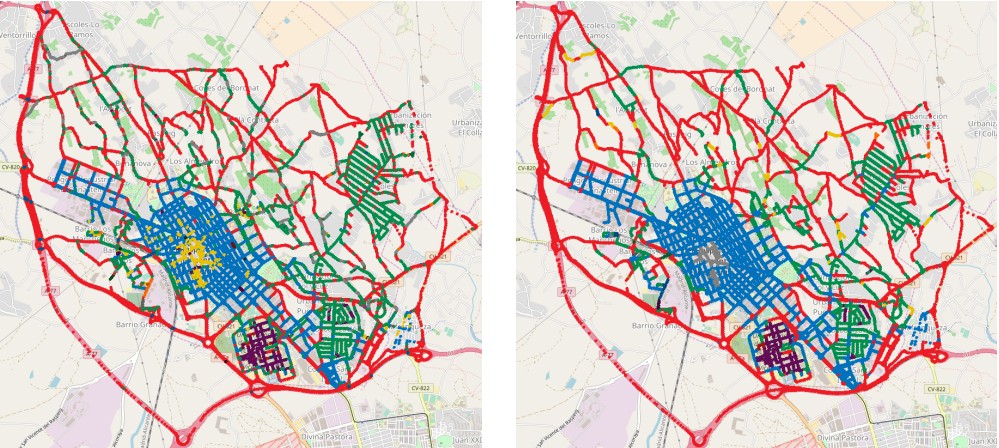

**Figure 20.** Comparison of smoothing results before (**left**) and after (**right**) applying KNN algorithm.

The original idea for executing a smoothing procedure was to decrease the uncertainty of the map by taking into account the categories assigned to the points in a zone; in other words, to create zones with a predominant label and fewer outlier points. Figure 21 shows a zoom-in a zone in the map before (left) and after (right) the smoothing procedure. In this case, the area corresponds to a residential, pedestrian location on the map. The main changes in the map after the smoothing procedure are marked with a red circle. In the left-hand image, the original map, the yellow points correspond to a pedestrian zone and the blue ones to a residential zone. It is notable that, in the limits of these zones, some streets appear that include points of both colors mixed together, for example, a yellow point between three or four blue points. Then, in the right-hand image, the original clusters keep their name and the pedestrian zone only changes color to gray. In this example, we can see that the streets in the borders of the zones now have only one predominant color. Therefore, the smoothing procedure helps to reduce flaws that could emerge in the original semantic mapping procedure.

Figure 22 shows another comparative example of the results generated by the smoothing procedure. In this example, it can easily be observed that points in "Villa Franqueza" street (the diagonal one on the map, marked in red) were grouped in the same cluster, the blue one representing the majority of the points in the zone, correcting the outliers generated in the map.

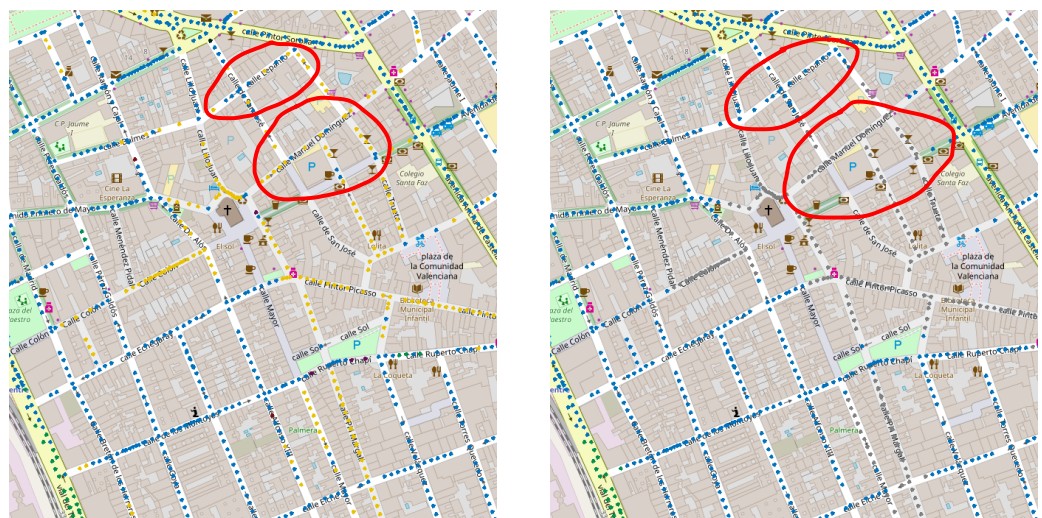

**Figure 21.** Comparison of smoothing results before (**left**) and after (**right**) applying KNN algorithm.

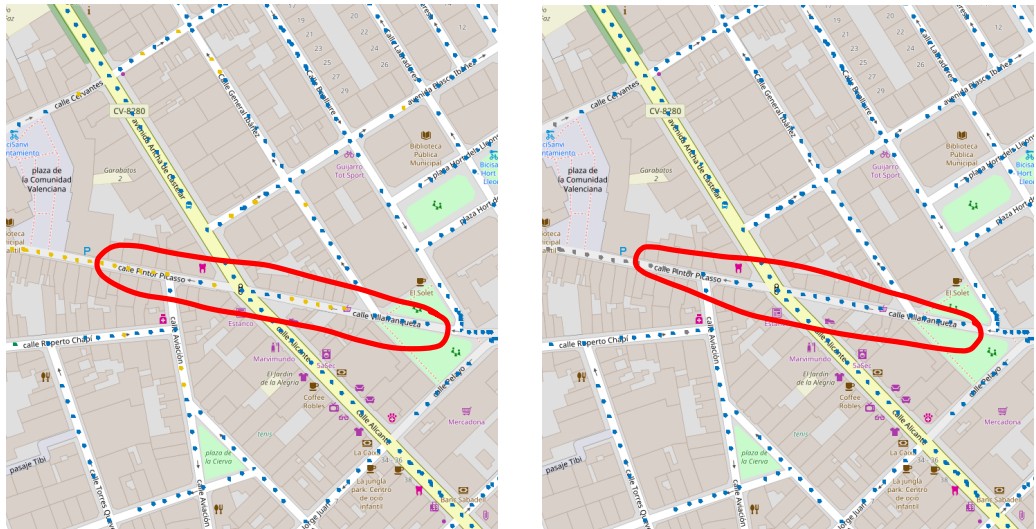

**Figure 22.** Comparison of smoothing results before (**left**) and after (**right**) applying KNN algorithm. A red circle indicates changes detected on images.

### 6.4. Ground Truth Comparison

Figure 23 shows a Google Earth Map capture of San Vicente del Raspeig, with the main zones that could be identified for this city and the respective roles of these zones. Table 4 shows images belonging to the zones identified in the city. Figure 24 compares the actual city map with one generated by the proposed method and smoothed using KNN. Here, we can appreciate that most of the generated clusters are related to actual regions of the city. The cluster in blue in the built semantic map includes the urban area and a great part of the industrial zone of the city. This can be understood due to similarities between the streets in both zones. In the same way, Figure 25 shows a comparison of the generated map with a segmentation produced by two open map platforms, namely, Google Maps and Open Street Maps. We can see some similarities between the three maps and also that our results are able to generate zones that are inside others, as well to produce new levels of separation between the areas in the map. Figure 26 shows images belonging to places that share similarities.

**Table 4.** Example images belonging to the ground truth regions.

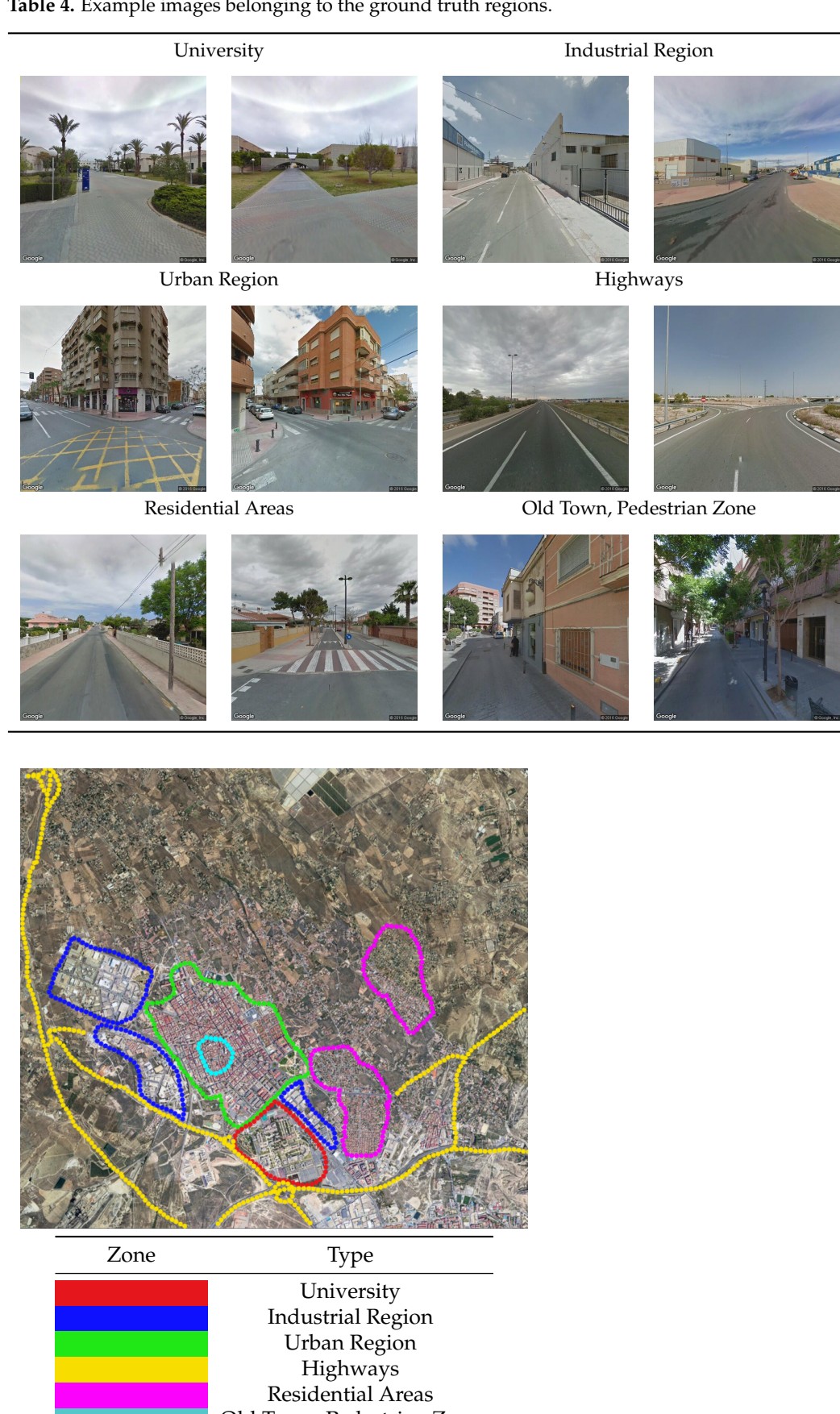

**Figure 23.** Ground truth for actual areas of the city selected for the study.

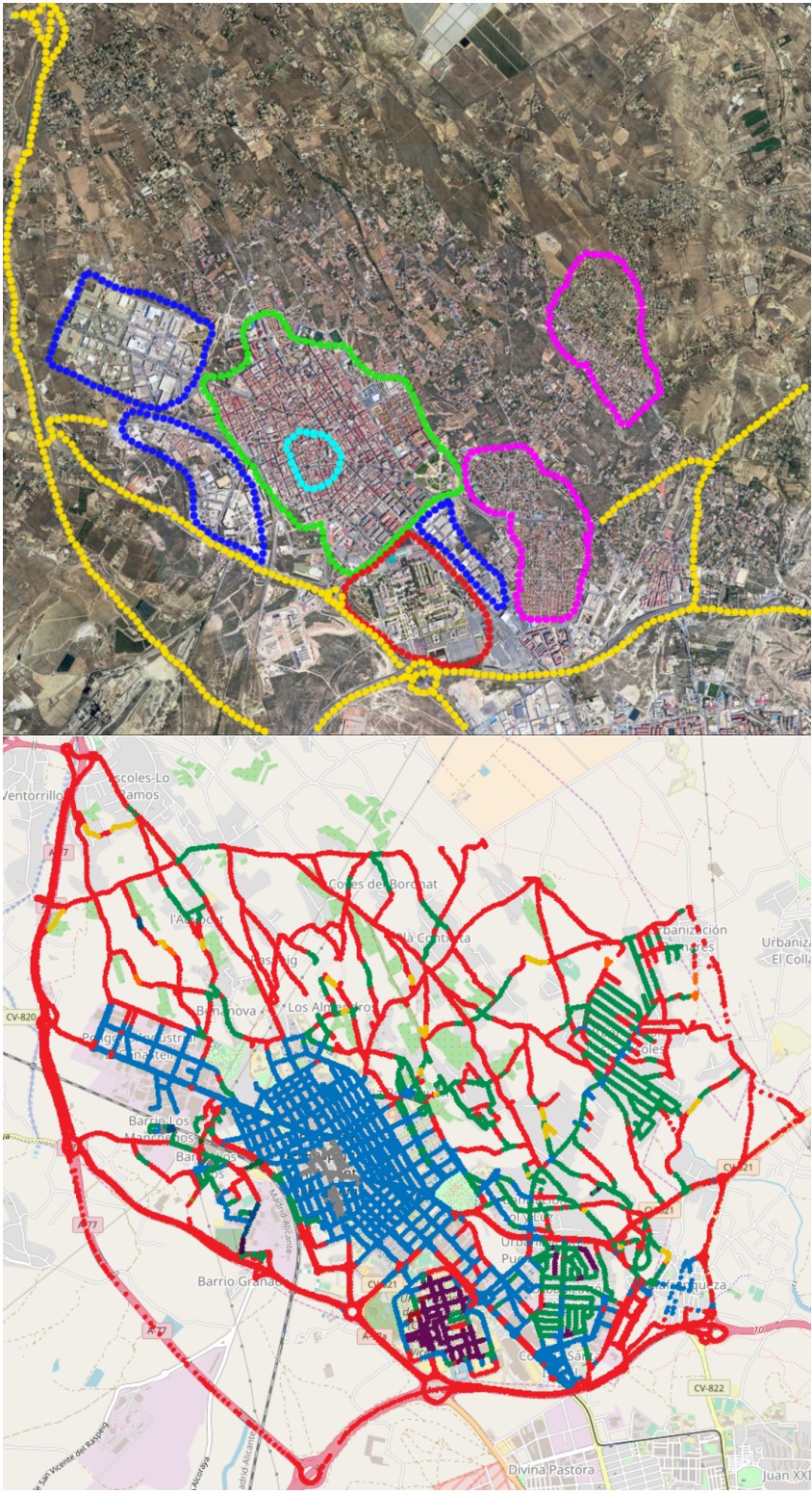

**Figure 24.** Comparison of results with the ground truth (**top**) and the smoothed map (**bottom**).

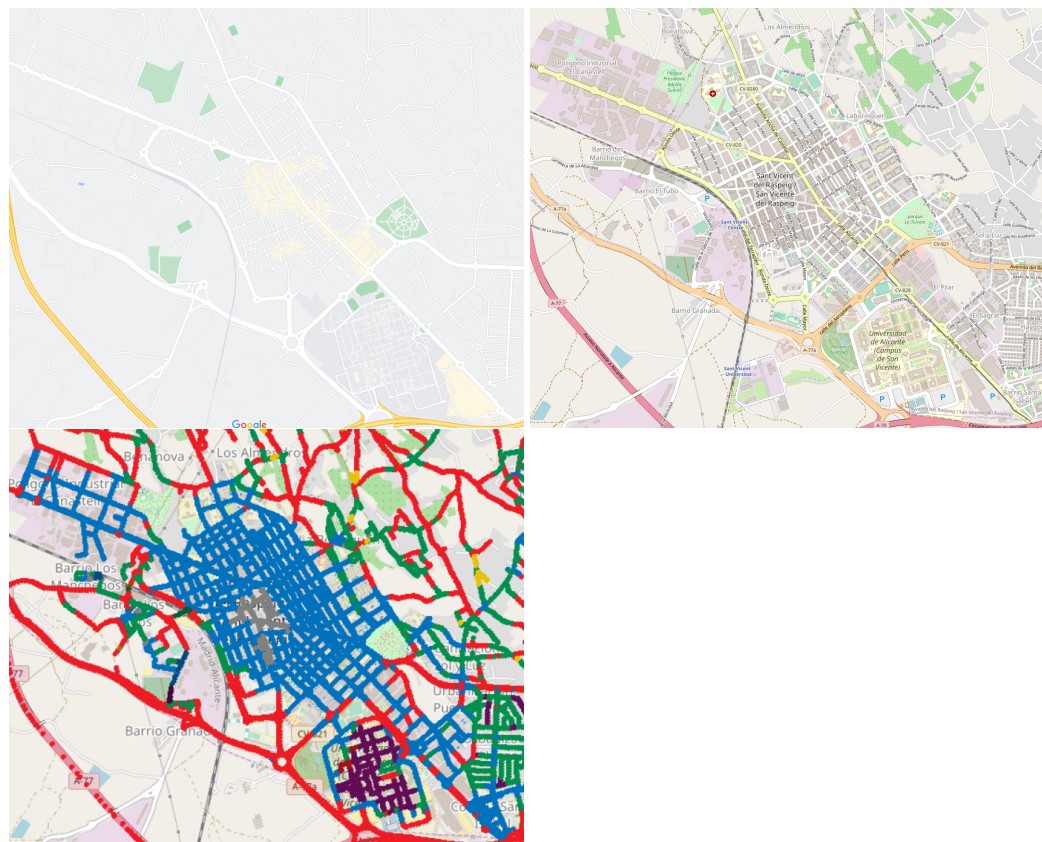

**Figure 25.** Comparison of results with zones generated in the city by Google Maps (**top left**) and Open Street Maps (**top right**), and the proposed method (**bottom**). These images focus on the center of the city.

Urban Region                                                    Industrial Region

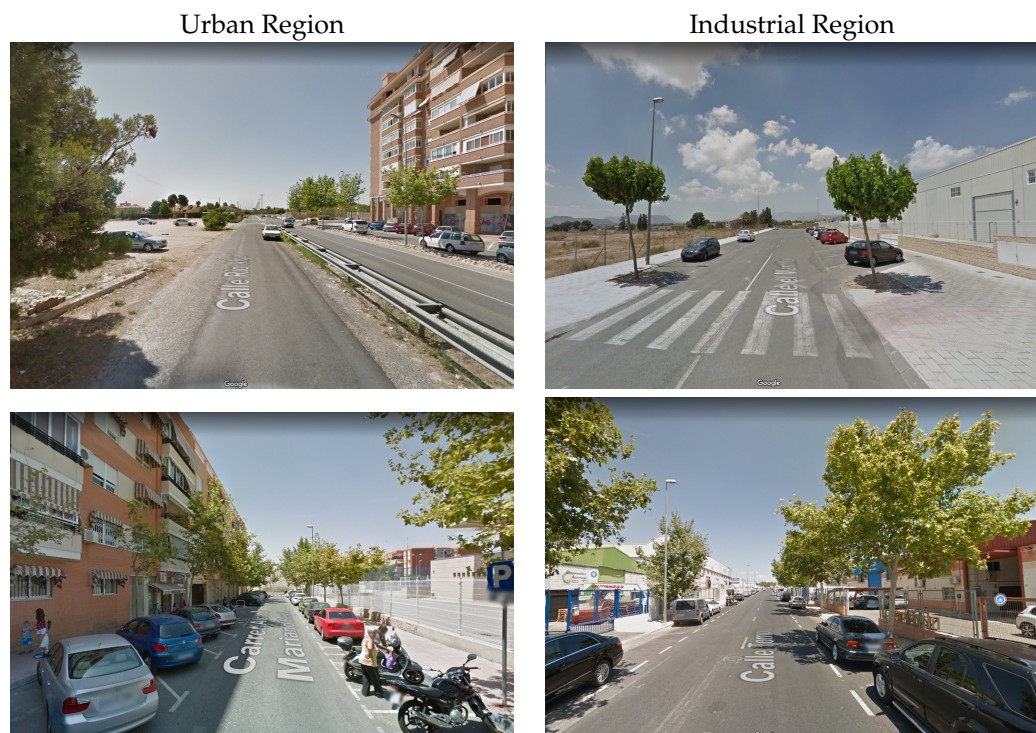

**Figure 26.** Similar images belonging to different areas in the city that were clustered together. The left-hand images were captured in an urban region of the city, whereas the right-hand images were obtained in the industrial area.

Furthermore, the method was able to merge together regions that are physically separated, as in the case of residential areas of the city. It is also worth noting the identification of pedestrian zones in the town center and the university area. The vast majority of images belonging to the highways in the city were also accurately merged and labeled, as appears in the ground truth map. Therefore, the representation is able to encode the zones present in the city.

## 7. Conclusions

Under our approach, the construction of semantic maps of a city can be achieved using images acquired from external tools such as Google Street View, avoiding the necessity of traveling around the entire city capturing the images. These images are of sufficient quality for use with CNN algorithms.

The semantic maps generated with the proposal are capable of grouping and identifying similar images of the city under the study. The zones generated by the algorithm and the semantic label assigned correspond to the actual function of the locations. The semantic descriptors then allow us to understand the possible role for the different points of the city.

The method proposed shows the capacity of separating very similar zones that accomplish different functions. These zones are very close to each other or within one another, and the method and the descriptors thus accurately encode the inner features of the region of the city. The distribution of the images in the clusters allows us to determine the maps that best represent the city. Moreover, the smoothing procedure helps to create meaningful maps with well-defined borders of zones.

The use of GPS coordinates allows the map to be used to define area policies for the city, based on the current content of the image and in the areas of the city and avoiding long periods of data-capturing in the city. Furthermore, an intelligent vehicle could use this system to acquire the semantic knowledge of the area in which it is moving. In doing so, the driver could know, for example, whether they must drive carefully when moving in a residential area or could obtain directions to a commercial area.

As future work, we will train a CNN model, using a reduced number of categories that only represent outdoor places in a city. The acquisition of the GPS points could also be addressed by means of specialized tools for navigation.

**Author Contributions:** Conceptualization, M.C. and J.C.R.; methodology, M.C. and J.C.R.; software, M.C. and J.C.R.; validation, J.C.R. and E.C.; formal analysis, J.C.R. and E.C.; investigation, J.C.R. and M.C. and E.C.; resources, M.C. and J.C.R.; data curation, J.C.R.; writing—original draft preparation, J.C.R. and M.C. and E.C.; writing—review and editing, J.C.R. and M.C. and E.C.; visualization, J.C.R.; supervision, M.C.; project administration, M.C.; funding acquisition, M.C. All authors have read and agreed to the published version of the manuscript.

**Funding:** This work has been supported by the Spanish Grant PID2019-104818RB-I00 funded by MCIN/AEI/10.13039/501100011033 and by "ERDF A way of making Europe". José Carlos Rangel and Edmanuel Cruz were supported by the Sistema Nacional de Investigación (SNI) of SENA-CYT, Panama.

**Institutional Review Board Statement:** Not applicable.

**Informed Consent Statement:** Not applicable.

**Data Availability Statement:** The data used to support the findings of this study are available from the corresponding author upon reasonable request. Also, these images can be found on Street View Static API (https://developers.google.com/maps/documentation/streetview/intro (accessed on 3 February 2022))

**Acknowledgments:** The authors acknowledge administrative support provided by Universidad Tecnológica de Panamá (UTP) and University of Alicante.

**Conflicts of Interest:** The authors declare no conflicts of interest.

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
