# Peer review of "Automatic Understanding and Mapping of Regions in Cities Using Google Street View Images"

_applsci, doi:10.3390/app12062971_

Round 1
Reviewer 1 Report
The paper presents research that results from other studies previously developed by the authors. In the introduction section, the authors cite only the works produced by the team and review state of the art in Section 2. It would be more convenient for a scientific article if the authors addressed other studies in the introduction to describe the background. Minor errors were detected in line 520, where the top would be left and the bottom would be right. I recommend that the authors review the numbering of the images and place them sequentially and that extras be numbered as an appendix. It is recommended that authors also check all the writing; minor typos were detected.
The paper makes a significant contribution to the semantic segmentation of cities based on images from digital mapping platforms.
Author Response
---The paper presents research that results from other studies previously developed by the authors. In the introduction section, the authors cite only the works produced by the team and review state of the art in Section 2. It would be more convenient for a scientific article if the authors addressed other studies in the introduction to describe the background.
The introduction has been improved taking reviewers recommendations.
----Minor errors were detected in line 520, where the top would be left and the bottom would be right.
We have assigned the actual position of the images in the text.
----I recommend that the authors review the numbering of the images and place them sequentially and that extras be numbered as an appendix.
Images’ size and latex template make a bit harder place figures in the more appropriate position, we have order some figures for improving the reading.
----It is recommended that authors also check all the writing; minor typos were detected.
The English has been corrected by a native speaker reviewer.
The article / paper has been proofread by a professional proofreader.
----The paper makes a significant contribution to the semantic segmentation of cities based on images from digital mapping platforms.
We would like to thank reviewer by their valuable comments.
Reviewer 2 Report
The author provides a new approach to generate a map with semantic information for each area of the city. The proposed method can automatically assign a semantic label for the cluster on the map. The results show the robustness of the proposed pipeline and the advantages of using Google Street View images, semantic descriptors and machine learning algorithms to generate semantic maps of outdoor places. In this article, the author clearly describes the proposal, methods and implementation procedure, it is exploratory in method.
Suggestions:
- Google Street View (Google Street View) usually has a long update cycle. This article is intended to provide route planning and convenience for outdoor travel. If the data is updated for too long, it may be biased. If possible, compare with the real road web.
- Some phrases in the text still need refinement. For example, Line 61-64, 204-206 have repeated
- Line65: Furthermore, themethod is unsupervised. CNN is supervised classification. Although the authors selected the MIT-trained model, it is still essentially supervised classification. The subsequent clustering algorithm is unsupervised classification. Is it appropriate to consider the method to unsupervised classification?
- In addition, each city will have landmarks. For example, the Forbidden City in China, the White House in the USA, etc. The sample library might have to be updated in a targeted manner.
- There is no problem with the authors' methodology and technical approach, which is complementary to the Google Map refinement. For the semantic segmentation, the authors chose four angular GPS point perimeter images of 0, 90, 180 and 270°. Why not use high-resolution remote sensing images for semantic segmentation and annotation? Combine the 4 angular images as a complementary refinement of the road network. This might be less complicated and less work.
- The section 3. KNN Smoothing comparisonincludes several comparison figures of smoothing results before (left) and after (right) applying KNN algorithm. But it is difficult to find obvious differences, the author could mark the corresponding position on the map(such as figure 21,22).
- Some statements about figure need to be check, For example, line510 and line520 mention the top and bottom.
Author Response
The author provides a new approach to generate a map with semantic information for each area of the city. The proposed method can automatically assign a semantic label for the cluster on the map. The results show the robustness of the proposed pipeline and the advantages of using Google Street View images, semantic descriptors and machine learning algorithms to generate semantic maps of outdoor places. In this article, the author clearly describes the proposal, methods and implementation procedure, it is exploratory in method.
Suggestions:
- Google Street View (Google Street View) usually has a long update cycle. This article is intended to provide route planning and convenience for outdoor travel. If the data is updated for too long, it may be biased. If possible, compare with the real road web.
Due to time limitations, we cannot make a more exhaustive comparison, but we already include a comparison with real road in the article (Figure 24 and 25).
- Some phrases in the text still need refinement. For example, Line 61-64, 204-206 have repeated
The English has been corrected by an English reviewer.
The article / paper has been proofread by a professional proofreader.
- Line65: Furthermore, the method is unsupervised. CNN is supervised classification. Although the authors selected the MIT-trained model, it is still essentially supervised classification. The subsequent clustering algorithm is unsupervised classification. Is it appropriate to consider the method to unsupervised classification?
Furthermore, the method is unsupervised: we do not include the categories a city has, but the method is able to obtain them using information from images.
The phrase has been changed by:
Furthermore, the method is semi-supervised: we do not define the categories a city has, but the method is able to obtain them using information from images.
- In addition, each city will have landmarks. For example, the Forbidden City in China, the White House in the USA, etc. The sample library might have to be updated in a targeted manner.
This is true, but in or case the target labels involve fewer specific labels such as, park or neighborhood. But the use of specific city landmarks can be included in the classification model to precisely identify a place in the city using the local terminology.
- There is no problem with the authors' methodology and technical approach, which is complementary to the Google Map refinement. For the semantic segmentation, the authors chose four angular GPS point perimeter images of 0, 90, 180 and 270°. Why not use high-resolution remote sensing images for semantic segmentation and annotation? Combine the 4 angular images as a complementary refinement of the road network. This might be less complicated and less work.
In this case the use of 4 regular images is due to the input size of the network. This requires a 640*480 image, then all the input images will be resized to these dimensions, loosing then some information in this procedure. But these suggestions could be another approach to evaluate as a future work.
- The section 3. KNN Smoothing comparison includes several comparison figures of smoothing results before (left) and after (right) applying KNN algorithm. But it is difficult to find obvious differences, the author could mark the corresponding position on the map (such as figure 21,22).
We marked the areas where these changes appear.
- Some statements about figure need to be check, For example, line 510 and line520 mention the top and bottom.
We have assigned the actual position of the images in the text.
We would like to thank the reviewer by their valuable comments.
Reviewer 3 Report
The paper presents a pipeline process aimed to generate a map having semantic information for different areas of a municipality, basing on Google services and CNN technique. The paper is very interesting, well written and presented and all aspects are thorough treated throughout the manuscript. Few comments are provided in the attached pdf that can be used for improving the manuscript.

Author Response
The paper presents a pipeline process aimed to generate a map having semantic information for different areas of a municipality, basing on Google services and CNN technique. The paper is very interesting, well written and presented and all aspects are thorough treated throughout the manuscript. Few comments are provided in the attached pdf that can be used for improving the manuscript.
Suggestions have been applied and answer is presented in pdf file.
We would like to thank the reviewer by their valuable comments.
We attached a pdf with the response.
-- To explicit
We have changed the redaction in abstract.
-- Here, I suggest to enlarge this part accounting for several ways to use Google images for developing machine learning techniques, e.g.,
https://doi.org/10.1016/j.autcon.2021.103936
https://doi.org/10.3390/data7010004
Thanks for the information. The new references have been added to the document.
-- here reference is missing
The reference has been corrected
--can you insert in the dendogram the used features to cluster?
In this case due to the dataset size the generated dendogram will be dificult to understand. The vertical axis would have almost 25000 points representing the points in the dataset.
-- why did you use only one CNN typology?
In this case we took advantage of a pre-trained CNN model, using the places dataset because it fits our objective. Other pre-trained model does not achieve as good results as the selected one., due to the categories correspond to objects.
-- What do you expect by changing the CNN?
Changing the model could allow us to use less labels and less dimensions for the semantic descriptor. These labels could be reduced to a set that fit the city. Then, map will be more accurate with the city.
